evolution/biophysics/genetics

evolution, Z-DNA, triplex, quadruplex, flipon, ALU

**Author for correspondence:**
Alan Herbert
e-mail: alan.herbert@insideoutbio.com

# ALU non-B-DNA conformations, flipons, binary codes and evolution

## Alan Herbert

Discovery, InsideOutBio, Inc., 42 8th Street, Unit 3412, Charlestown, MA 02129, USA

AH, 0000-0002-0093-1572

ALUs contribute to genetic diversity by altering DNA's linear sequence through retrotransposition, recombination and repair. ALUs also have the potential to form alternative non-B-DNA conformations such as Z-DNA, triplexes and quadruplexes that alter the read-out of information from the genome. I suggest here these structures enable the rapid reprogramming of cellular pathways to offset DNA damage and regulate inflammation. The experimental data supporting this form of genetic encoding is presented. ALU sequence motifs that form non-B-DNA conformations under physiological conditions are called flipons. Flipons are binary switches. They are dissipative structures that trade energy for information. By efficiently targeting cellular machines to active genes, flipons expand the repertoire of RNAs compiled from a gene. Their action greatly increases the informational capacity of linearly encoded genomes. Flipons are programmable by epigenetic modification, synchronizing cellular events by altering both chromatin state and nucleosome phasing. Different classes of flipon exist. Z-flipons are based on Z-DNA and modify the transcripts compiled from a gene. T-flipons are based on triplexes and localize non-coding RNAs that direct the assembly of cellular machines. G-flipons are based on G-quadruplexes and sense DNA damage, then trigger the appropriate protective responses. Flipon conformation is dynamic, changing with context. When frozen in one state, flipons often cause disease. The propagation of flipons throughout the genome by ALU elements represents a novel evolutionary innovation that allows for rapid change. Each ALU insertion creates variability by extracting a different set of information from the neighbourhood in which it lands. By elaborating on already successful adaptations, the newly compiled transcripts work with the old to enhance survival. Systems that optimize flipon settings through learning can adapt faster than with other forms of evolution. They avoid the risk of relying on random and irreversible codon rewrites.

# 1. Introduction

The small 140–300 base-pair (bp) ALU size belittles the extent of evolutionary improvisations enabled in Hominidae by this class of short interspersed nuclear element (SINE) repeat family. They are named for the restriction enzyme Alu I from *Arthrobacter luteus* that cuts within them, producing short fragments when two ALU elements are nearby. ALUs are derived from the non-coding 7SL RNA of the signal recognition protein. They spread invasively by retrotransposition, with each attack initiated by a different family, generating over one million copies to occupy almost 11% of the human genome [1,2]. The most efficient transposition was by ALUs that are dimeric, composed of a left and right arm derived from 7SLRNA joined by a deoxyadenosine-rich spacer and with a 3′ polydeoxyadenosine tail of variable length (figure 1*a*). Like other DNA, ALUs are packaged into nucleosomes, wrapped around histone octamers. DNase hypersensitive sites are present every 10.2 bp, where the energetic cost of DNA bending is lowest. The flexing of d(CG) dinucleotides, which repeat every 32 nucleotides, contributes to this pattern [9]. The core nucleosome footprints from −80 to +69 of ALU with the spacer that separates the Alu monomers positioned in the linker region between each nucleosome pair [10,11]. The stiffness (i.e. resistance to bending) of the adenosine-rich ALU spacer sequence contributes to their exclusion from nucleosomes [12].

# 2. Why ALUs are interesting?

There are many reasons to consider ALUs of interest, reflecting the many different factors driving their evolution. First, ALUs have played a very large role in shaping the human genome [13]. They are widely dispersed and mostly found in RNA polymerase II (RP2) transcribed genes where they can act as signals for splicing and transcriptional termination [2,14]. Only a minority (0.2%) of ALUs undergo exonization, often with deleterious effects. Around 86% of ALU exons are inserted in the reverse orientation and contain an in-frame stop codon or a frameshift resulting in loss of protein function [15]. Second, the spread of ALUs through the genome during evolution offers a rapid way to alter the read-out of genetic information, independently of codon mutation. They are bound by a wide range of transcription factors [16] that enable the co-regulation of their host genes, even when those genes are present on different chromosomes [17]. Third, the extent of ALU transcription allows cells to monitor their health. High levels of ALU RNAs are sensed and initiate inflammatory responses leading to disease [18]. Mechanisms such as RNA editing [19–21], originally tasked with stopping the existential threat posed by ALU spread throughout the genome [4], now regulate innate immunity [22]. Many cancers exploit these pathways to remain immunologically silent [23–26]. Fourth, as will be reviewed here, the existing data are consistent with a model where the non-B structures (NoBs) ALUs form enable dynamic responses to environmental changes. In this synthesis, ALU sequences vary genomic read-out by changing conformation, acting as programmable switches to alter the transcripts produced. Once the invader, ALU elements now enhance the evolution of their host.

# 3. What does ALU RNA do?

ALU RNAs have roles that vary by how they are transcribed. A minority of ALUs (1.5%) undergo transcription by RNA polymerase III (RP3), using the same A and B boxes as 7SL RNA (figure 1) [14]. Many RP3 transcripts are expressed in a tissue-specific manner. An analysis of 27 cell lines, 16 primary cell types, and 45 tissues using a recently developed high-stringency method for identifying RP3 transcripts revealed that their cell-specific expression is predicted by chromatin modification, not by differences in ALU sequence [14]. Transcription is highest for ALU elements that lie within 10 000 bases of genes transcribed by RP2. The RP2 and RP3 transcription units cooperatively induce local chromosomal loop formation [27,28]. Some RP3 units produce an enhancer RNA (eRNA) that guides the assembly of transcription factories [29,30]. Other RP3 transcripts act in *trans* to inhibit RP2, with the ALU right arm binding to the RP2 cleft [31]. ALU self-cleavage relieves this inhibition during heat shock. The ALU-Y ribozyme cut site is at position 51 of the left arm. Cleavage is enhanced by the enhancer of zeste 2 polycomb repressive complex 2 subunit (EZH2), even with catalytically inactive versions of the protein [32,33]. ALUs can inhibit translation by inhibiting ribosome assembly [34,35]. In addition, they induce phosphorylation of PKR (double-stranded RNA (dsRNA) activated protein kinase encoded by EIF2AK2) to inhibit translation and initiate stress granule formation [36].

RP2 ALU transcripts have different roles, acting in *cis* to alter RNA processing and turnover [37]. The ALU elements stimulate alternative splicing. They enable the production of circular RNAs (cRNAs),

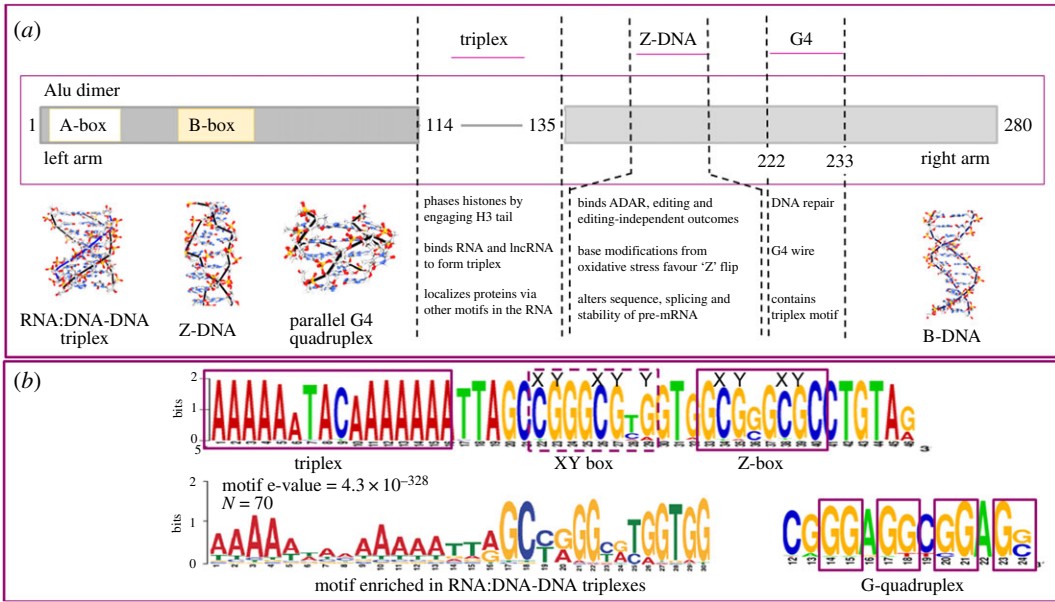

**Figure 1.** ALU sequences, non-B-DNA conformations and modifications. (*a*) An Alu dimer consisting of left and right arms with a triplex forming adenosine-rich spacer. The A- and B-boxes promote transcription by RNA polymerase III. Triplex [3] and Z-DNA formation [4] dynamically change the information read-out from the genome, making different outcomes possible. ALUs can also potentially stack two-layered intramolecular quadruplexes to form G4Q wires [5]. DNA models are displayed from the following sources: triplex model (PDB:1RX3) with the third strand in blue, Z-DNA (PDB:4LB5), a parallel strand G4Q (PDB: 1NYD), B-DNA (PDB:1BNA). (*b*) Sequence Weblogos (generated at https://weblogo.berkeley.edu/logo.cgi). The upper logo is from the consensus ALU sequences with canonical triplex and Z-forming sequences boxed by solid lines (table 1 in [1]). The dotted box contains sequences where base modifications decrease the energetic cost of flipping their conformation from right- to the left-handed DNA. Cytidine residues with the potential to undergo methylation and further oxidation are marked with an 'X' [6] while those favouring oxidation of guanosine to form 8-oxo-dG are marked with 'Y' [7], resulting in an 'XY' DNA sensor for oxidative stress. The lower left logo summarizes experimentally determined, triplex-forming ALU sequences. The difference between this sequence and the consensus sequence reflects the evolutionary adaptations that improve targeting of RNA strands to specific ALU duplexes [8]. The lower right logo captures the sequence reported to form an intramolecular quadruplex with the four G quartet forming residues boxed [5].

which act as sponges for microRNA to alter phenotype [38]. RP2 also transcribes short (100–300 bp) intronic ALUs that enhance phase transitions leading to nucleolus formation [39,40] and sets of AluACA transcripts with unknown function [41]. ALUs form double-stranded RNAs (dsRNA) when inverted repeat elements (AIRE) are close enough to base-pair with each other. AIRE localize messenger RNAs to nuclear paraspeckles [42] and cytoplasmic stress granules [43]. They are substrates for dsRNA editing [19–21]. The different ALU roles reflects an evolutionary history with layers upon layers of adaptations. A similar evolutionary complexity is attached to ALU NoBs.

# 4. What are non-B-DNA structures?

Here I discuss Z-DNA, triplexes and quadruplexes (figure 1), referencing the work of many talented scientists. Z-DNA duplexes are twisted to the left rather than the right as in B-DNA. Z-DNA forms directly from B-DNA by flipping base-pairs over, causing the strands connecting the bases to zig-zag to the left. Triplexes are right-handed with a third strand wrapped around the two strands of B-DNA. Quadruplexes are usually right-handed with four strands of DNA wrapped around each other. The most studied quadruplexes are built from a quartet of guanosine hydrogen-bonded to each other (G4Q). Here I focus on their role in the biology of ALUs, trying as we proceed to separate evidence from *in vitro*, *in vivo* and *in silico* studies from other sources. The reader is reminded that this field is rapidly evolving with many recently developed technologies paving the way to new discoveries. The discussion is intended as a guide to future experimentation.

# 5. Why are non-B-DNA structures interesting?

Alternative DNA structures epitomize a non-traditional way of encoding genetic information [44,45]. I refer to the sequences that dynamically switch conformation under physiological conditions as flipons [44]. Flipons trade energy for information [44]. They are dissipative structures [46], requiring work to change their conformation. The flip enables targeting of the cellular machinery to active genes instead of to other sequences. By localizing RNA processing complexes, flipons alter the transcripts compiled from a gene. The RNAs produced depend on context and change with flipon conformation.

Much of this discussion is an interpretive review of the literature, intended to highlight what we don't know and where we need to explore. The presentation starts with the following question. If NoBs are so important *in vivo*, why is this entire field still controversial? Many readers will be aware of the view that NoBs lack biological relevance. Indeed NoBs were long ago penned into history as a biological dead end, at least for Z-DNA, first by the professional journal editors [47] and then by the philosophers [48]. The opinion is largely based on the belief that absence of proof is equivalent to proof of absence [49], the famous black swan fallacy. Each to their own measure of success, but the case for the biological relevance of NoBs is now very strong.

# 6. Why is the science of non-B-DNA structures difficult?

The study of NoBs has always been hard because of the many different sources of potential experimental bias. The question with NoBs experiments is always the same: are the findings valid or just an artefact? Experiments on NoBs are easily confounded. NoBs can form *in vitro* during procedures designed to show their presence *in vivo*. Procedures may inadvertently release bound histones and generate enough negative supercoiling (NSC) to either flip right-handed DNA to the left-handed conformation or to create single-stranded regions that fold as G4Q [50,51]. An example of this effect is the binding of Z-DNA-specific antibodies to Drosophila polytene chromosomes. Results depend on fixation conditions [52]. Similar problems are reported for antibody staining of G4Q [51], which are remarkably stable under a wide range of conditions [53]. With *in vivo* approaches, introduction of a Z-DNA binding protein or a G4Q binding drug into a cell may stabilize the conformation in an unphysiological manner. In some NoBs studies, artefacts only become known with time. One example is the co-immunoprecipitation of RNA complexes that form by rapid exchange of RNAs between proteins only after cell lysis [54]. Another is the preferential enrichment by ChIP-Seq, regardless of the antibody used, of certain sequences. The problem was first identified through the efforts of the Encode Project and summarized by a sequence blacklist that include examples like centromeric repeats [55]. Earlier results based on this technique that relate to NoBs must therefore be interpreted cautiously [56]. There is also a reagent bias. For example, permanganate only modifies thymine bases [57] and kethoxal only reacts with guanosine bases [58]. Together they provide a complementary approach for footprinting single-stranded regions associated with NoBs *in vivo* but individually may miss certain sequences. A different type of bias arises from data missing due to experimental design: it is routine to exclude from genome-wide analyses sequences like ALU. They are removed from analysis of d(CpG) modifications by protocol [59] and also from some *in vivo* footprinting assay results [45]. Also heavily edited ALUs are often discarded without analysis as they do not map to the genome. The proper pipeline is needed to study such sequences [60]. To increase our understanding of flipon biology, we need to use different data collection protocols.

The biochemical identification of NoBs binding proteins is also challenging and requires rigorous controls with structural studies to confirm findings. For Z-DNA, there have been many solo publications in which conformation-specific proteins are claimed, but where verification by biochemical and structural studies is absent [61–63]. Subsequent studies often show these proteins to have a different functions altogether, unrelated to NoBs. One example is Zuotin, a ribosomal co-chaperone for nascent peptides originally proposed as a Z-DNA binding protein [64]. The same is true for the claim that most Z-DNA binding proteins are actually lipid binding proteins [65]. Even when good reagents are available against proven Z-DNA binding proteins, like the p150 isoform of the adenosine deaminase RNA-specific enzyme ADAR, incorrect statements echo through the literature. The finding that protein p150 location in a limited number of tumour cells is cytoplasmic after interferon induction has been highly cited [66], despite the clear nuclear localization of p150 demonstrated in the original publication [66]. The presence of p150 in both the cytosol and the nucleus is consistent with the nucleocytoplasmic shuttling of p150 [67–69]. Recent surveys of normal

tissues show that p150 is mostly nuclear in murine thymus, spleen and prefrontal cortex [70–72]. Conversely, the cytoplasmic localization during stress of ADAR p110 isoform that lacks the Z-DNA binding domain, is not often referenced [73].

Additional challenges to identifying conformation-specific non-B binding proteins also exist. Proteins like the high-mobility group family are disordered random coils in solution and only fold on binding DNA. They engage a variety of NoBs, including triplexes and quadruplexes, none specifically [74,75]. Another issue arises with sequences that adopt more than one NoB, raising the question as to which interaction is biologically relevant. For example, the telomeric sequences that form G4Q *in vitro* stabilize a triplex *in vivo* in which the G-strand overhang folds back onto a telomere proximal sequence to form a T-loop [76,77]. There is also debate as to whether the relevant NoBs are formed from DNA, RNA or a hybrid. The direct answer is that all three classes of NoB duplexes can serve an informational role. The better response is to focus on the transient nature of the NoBs formed *in vivo*. They allow a cell to switch from one phenotypic state to another. Their persistence is often associated with disease.

The work on Z-DNA satisfies the necessary criteria for demonstrating the *in vivo* relevance of NoBs: *in vitro* biochemical validation to show function; angstrom resolution structures to eliminate ersatz explanations; *in vivo* studies to confirm mechanism; Mendelian genetics to confirm disease causality. The development of well-controlled assays [78] enabled the characterization of the Zα domain in the double-stranded RNA editing enzyme ADAR p150 isoform, then the identification of a host of homologues. ADAR Zα binds specifically to Z-DNA without sequence preference [79–81]. The initial biochemical characterization demonstrating structure-specific binding [81] was confirmed by NMR and crystallographic studies [79,82,83]. Mendelian genetics and biochemical characterization of the disease-causing variants then established a biological role for the Z-duplex in the negative regulation of type I interferon responses [84]. Recent publications show that many cancers exploit this property to silence immune responses [26].

Further validation for a role for the Z-duplex in biology is provided by studies of the related protein, Z-DNA binding protein 1 (ZBP1, also known as DAI or DLM-1) [85–89]. ZBP1 has two Zα domains, the first of which is absent in many splice isoforms [90,91]. Deletion or mutation of Zα residues in ZBP1 confirm a central role for the Z-conformation in RIPK3-dependent necroptosis that occurs in inflammatory bowel disease (IBD), dermatitis and influenza virus infection. Mutations to Z-binding residues in the second Zα domain ameliorate these phenotypes [85–87,89,92]. E3L, a vaccinia virus Zα orthologue, was recently shown to inhibit ZBP1-induced necroptosis during infection [93]. ZBP1 plays other roles in other cell death pathways involving apoptosis and pyroptosis [92]. The interactions between ADAR p150 and ZBP1 and their specific roles requires further study. One possible generalization is that ZBP1 induces early inflammatory responses that are later negatively regulated by ADAR.

# 7. What is the connection between ALUs and flipons?

Zα-dependent RNA editing of ALU RNAs by ADAR provides evidence that Z-flipons regulate biological responses such as the type I interferon responses in innate immunity [4]. The finding raises many questions. Has natural selection exploited the spread through the genome of Z-flipons by ALU to regulate additional pathways? If so, is the same true for other classes of flipon? What is the evidence? We have only partial answers to these questions at this time. Not all the necessary experiments have been done and not all the relevant data collected. What follows is an interpretation of the data we have with many of the gaps highlighted as hypotheses that require experimental evaluation.

# 8. Z-flipons

ALU include sequences where the base-pairs flip over to form Z-DNA (Z-Box, figure 1*b*), a structure in which the double-helical backbone zags left instead of twisting right as it does in Watson–Crick B-DNA [4]. The energetic cost of the transition varies with the length and nucleotide composition of a DNA segment. Z-DNA formation is favoured with alternating pyrimidine, purine sequences such as d(CG) repeats [4]. Not all sequences can flip under physiological conditions, giving informational value to those that can. Predicted Z-DNA forming sites in the human genome are present at transcription start sites (TSS) of the annotated genes on chromosome 22 [94]. Z-DNA forms in the MYC promoter as measured by Z-DNA-specific antibodies [95] and stabilizes the open chromatin structure at the CSF1

promoter, a process dependent upon the BAF complex (BRG1- or HBRM-associated factors) [96]. A genome-wide *in silico* survey reveals that the majority of predicted Z-DNA forming elements lie in repeat elements, either as simple sequence repeats such as $d(AC)_n$ and $d(GT)_n$ or in the right arm of ALU dimers, with the propensity to form Z-DNA varying by ALU family (figure 1) [4].

The existence of proteins that bind Z-DNA with high affinity provide evidence for the biological relevance of this conformation. The best-known example involves the p150 isoform of ADAR (adenosine deaminase RNA-specific enzyme) with its Zα domain that binds Z-DNA in a structure-specific but sequence-independent manner [79,83,97], even to sequences like $d(AT)_n$ that do not form Z-DNA easily *in vitro* [81]. Mutations to the amino acid residues contacting Z-DNA, like N173S and P193A, are causal for the type I interferonopathies characteristic of Mendelian diseases, such as Aicardi–Goutières syndrome [84], revealing an important regulatory role for the Z-duplex in innate immunity. The P193A mutation is associated with reduced double-stranded RNA (dsRNA) editing [98] and increased induction of the interferon response factor 3 (IRF3) gene [99].

Studies have revealed that ADAR Zα binds a number of genomic sites. ChIP-Seq based on an ADAR Zα dimer identified a total of 391 Z-forming sequences, mostly (46%) in promoters and associated with active histone marks such as H3K4me3 and H3K9ac [100]. The mapped genes are enriched for replication-dependent histones prone to citrullination [101], cell cycle factors, transcription factors and topoisomerase 3A, which is essential for resolution of D-loop recombination intermediates [102]. The centromeric repeats found in an earlier study [56] are most likely Encode blacklist sequences [55], as they lack a strong Z-forming motif. A recent study of the ADAR Zα regulation of fear extinction in mice found that 80% (97/122) of RNA editing sites overlap with Adar1 DNA binding, with an overrepresentation of SINE/LINE elements nearby. Both DNA binding and RNA editing were absent when a loss-of-function Zα (N175A/Y179A) mutant was tested in the *in vivo* assay [70]. This response also depends upon prion formation by the cytoplasmic polyadenylation element binding protein CPEB3 [103]. Whether Z-formation affects memory by altering CPEB3 splicing or the processing of miRNA loci that regulate CPEB3 expression is unknown at present.

ZBP1 Zα-dependent functions have been investigated in a number of studies. In mouse models, ZBP1 activates necroptosis mediated by RIPK3–MLKL [86,89,92]. In these studies, tissue-specific gene knockouts were used to induce disease by preventing RIPK1/FADD complex-induced inactivation of ZBP1/RIPK3 kinase signalling through caspase 8 activation. Mouse models used epidermis-specific RIPK1 deficiency to produce skin inflammation and intestinal epithelial-specific FADD deficiency to induce colitis. The necroptosis produced was due to Z-binding by ZBP1 as either deletion or mutation of the Zα domains prevented disease. Activation of ZBP1 was by endogenous retroelement expression and dependent on the second Zα domain. In skin lesions, SINEs were upregulated and long terminal repeat elements downregulated relative to normal tissue [86]. Skin lesions were partially ameliorated by a combination of reverse-transcriptase inhibitors, suggesting that DNA/RNA hybrids can trigger ZBP1 [86]. Activation of ZBP1 by endogenous retroviruses (ERV) is also reported in human inflammatory bowel disease (IBD) patients, possibly due to low expression levels of the repressive methyltransferase SETDB1 [87]. In mouse models with selective *Setdb1* knockout in intestinal stem cells, increased levels of ERV in diseased tissue relative to normal was also present. In models of influenza viral infection, where the virus replicates in the nucleus, ZBP1 binds defective viral genomes in immunoprecipitation assays, although binding to ALU elements or other retroelements was not evaluated. Forced nuclear localization of ZBP1 in this model increased necroptosis [88]. Collectively these results reveal an important role for ZBP1 and endogenous flipons in regulating cell death.

What initiates a change in Z-flipons conformation? The unwinding of B-DNA powers the flip from B-DNA to Z-DNA. Enzymes generate the required energy as they work to transcribe RNA. Polymerases generate negative supercoils (NSC) faster than topoisomerases relax them [45,104–106]. The negative supercoiling they produce accumulates upstream of the polymerase and stabilizes the left-handed conformation [107]. NSC is highest at the TSS where Z-DNA forming sequences are found [94–96]. The topoisomerase I activity here is lower (as judged by use of camptothecin to trap catalytically active enzyme on DNA), even though levels of the enzyme are much higher than in gene bodies (as judged by ChIP-Seq) [108]. These effects are enhanced when upstream DNA is anchored to limit their dissipation along the helix. Anchoring can be either by scaffold attachment or by DNA: RNA hybrid formation [109]. The energy stored in NSC and Z-DNA is then available for assembly of transcriptional complexes and to load RNA polymerase at the promoter [94–96].

The energetic cost of Z-formation *in vitro* is lowered by polyamines such as spermine and spermidine, but not by acetylspermidine or putrescine [110]. The levels of spermine and spermidine vary during the cell cycle and increase during oxidative stress [111] with catabolism due to acetylation by spermidine/spermine

N1-acetyltransfease 1 (SAT1). Polyamines exist in cells as nuclear aggregates (NAPs) [112]. *In vitro*, NAPs can promote Z-formation and form higher order structures that wrap about DNA to form fibrils. The biological role of fibrils is not clear. One untested possibility involves a role in the production of neutrophil extracellular traps (NETs). NETs are made of chromatin structures from neutrophils extruded into the extracellular space as a result of pyroptosis, which is induced through ZBP1 interactions with RIPK1 and dependent on NFκB activation of the NLRP3 inflammasome pathway [113–115]. The NETs are stabilized by chlorinated polyamines and are pro-inflammatory [116]. *In principle*, NAPs could activate ZBP1 by promoting Z-DNA formation and lead to NET formation via gasdermin [117]. A similar outcome could account for a rare monogenic form of systemic lupus erythematosus (SLE), a disease in which anti-Z-DNA antibodies are frequent [118,119]. In two multiplex families, an X-linked recessive form of SLE was due to SAT1 loss of function variants that would be anticipated to increase spermine levels. In general, NET degradation is impaired in SLE, [120] and increases the risk of lupus nephritis [121].

The energetic cost of Z-formation *in vitro* is lowered by a number of base modifications. These include the 8-oxo-7,8-dihydro-2′-deoxyguanosine (commonly called 8-oxo-dG) adduct [7], which is produced by a non-enzymatic, chemical reaction during oxidative stress [122]. ALUs are particularly susceptible to formation of 8-oxo-dG due to the presence of many cytidine, guanosine dinucleotide steps that undergo rapid oxidation [123]. These steps also favour Z-DNA formation. The combination of sequence and 8-oxo-dG in the Z-box then enhances a flip to the Z-conformation (figure 1*b*). Interestingly, ALU elements in cells are highly susceptible to X-ray-induced oxidative damage, but in a survey of tumours, there is no increase in the incidence of single nucleotide variations compared to the flanking sequences (fig. 5i in [122]), suggesting that the ALU sequence containing the Z-DNA forming motif is constrained by function.

Modification of cytidine in the 5-position also alters the *in vitro* energetics of Z-DNA formation in an unexpected manner [6]. While 5-methylcytosine (5mC) is a well-known modification, base oxidation produces other derivatives of cytidine (C). These lesser known products arise during oxidative stress [124] and through the action of the ten-eleven-translocation (TET) dioxygenase protein family, which promotes the sequential formation from 5mC of 5-hydroxycytosine (5hC), 5-formylcytosine (5fC) and then 5-carboxylcytosine (5cC). The reactions occur in the order $C \rightarrow 5mC \rightarrow 5hC \rightarrow 5fC \rightarrow 5cC \rightarrow C$ (figure 1*b*) [125]. While 5mC promotes the Z-DNA flip *in vitro*, 5hC inhibits the transition, stabilizing the B-DNA flipon conformation instead [6,126]. The effect of 5fC and 5cC on Z-formation is interesting. If both cytosines in a CpG step are modified, there is a steric clash between the adducts that inhibits Z-formation [126] (figures 2*b* and 3*a*). If the modified cytosines are spaced to prevent the clash, the flip to Z-DNA is actually enhanced [6]. The Z-forming potential of a CpG island then varies with the density of methylation. At high levels of 5mC, flipons will probably be locked into the B-conformation by cytosine oxidation, due to steric clashes preventing Z-DNA formation. In hemi-methylated regions, clashes will no longer occur, with 5fC and 5cC now favouring the flip to Z-DNA. Hemi-methylated d(CpG) dinucleotides are found *in vivo*. They flank CCCTC-binding factor (CTCF) sites that anchor chromatin loops [127,128] and are maintained during replication [127]. The hemi-CpGs cluster together and are present only on one strand, with rotational symmetry about the CTCF motif. They are enriched in gene bodies, but not promoter regions, which are unmethylated [127]. A different way to relieve d(5fCpG) and d(5cCpG) steric classes is by transcription. The modifications then exist only on the DNA strand with the RNA unmodified in nascent transcripts. The cytosine adducts can then enhance Z-formation by the hybrid, and promote any Z-dependent RNA processing. The regulation of Z-flipon conformation *in vivo* by 5hC and its oxidative products needs experimental validation. There is indirect evidence for this possibility. The ratio of 5fC and 5cC to 5hC varies between different repeat elements, between young and old members of a repeat families and between different cell types, suggesting active control of this reaction pathway [129]. The variation depends in part on levels of the repair enzyme thymine DNA glycosylase (TDG), which excises 5fC and 5cC from DNA. The reaction is stimulated by the base excision repair enzyme NEIL1 (nei like DNA glycosylase 1), even when NEIL1 is catalytically dead. Currently it is believed that 5hC removal in animals requires its oxidation to 5fC [129,130].

Cytosine adducts potentially impact gene expression in a number of ways. They may directly influence placement of nucleosomes as 5fC forms a Schiff's base with H3 histone tails during the assembly of octamers [131]. Other effects of these modifications depend on their location in a gene. In the case of 5hC, experimentally measured transcription decreases when 5hC accumulates in the promoter region, but increases when 5hC accrues in the gene body (figure 2*c*) [132]. One testable explanation for the apparently contradictory effects on transcription involves the inhibition of Z-DNA formation by 5hC. At the promoter, low levels of 5hC probably favours Z-DNA formation, potentiating a role in enhancer and promoter assembly. In contrast, high levels of 5hC in the gene body potentially diminish inhibition of RP2 processivity by Z-DNA, thereby increasing transcription rate [4].

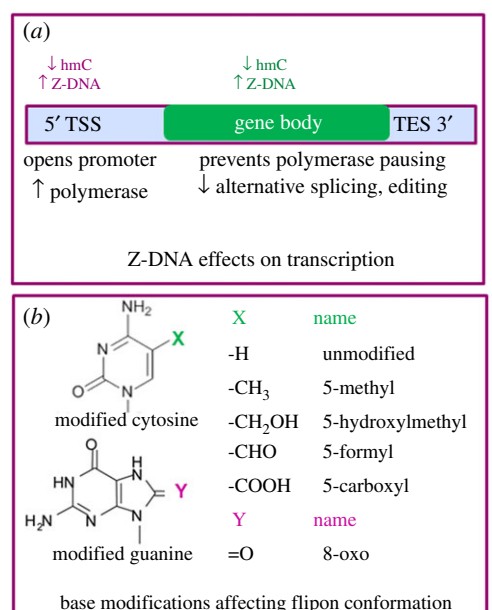

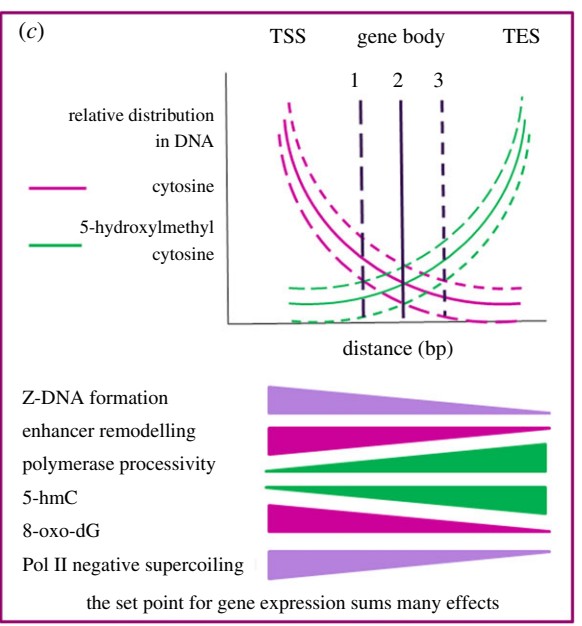

**Figure 2.** Z-DNA formation. (*a*) 5hC modifications lower Z-DNA formation by a flipon sequence. The consequences depend on whether 5hC forms in the promoter region where it inhibits promoter opening associated with Z-formation or in the gene body where it promotes RNA polymerase processivity by preventing Z-formation. (TSS, transcription start site; TES transcription end site) [132]. (*b*) The modifications to cytosine (labelled X as in figure 1) and guanine (labelled Y as in figure 1) described in the text. (*c*) Differences in negative supercoiling and base modifications, as indicated by the wedges, determine the optimal set points for expression of different messages from each gene. The conformation of Z-flipons within a gene body allows a dynamic change to the transcripts compiled from a gene. The vertical lines represent the best set points for three different genes (gene 1: long dash, gene 2: no dash, gene 3: short dash). The vertical lines (labelled by gene number) intersect the other two transcript-specific lines at the point where read-out is optimal for each gene, balancing enhancer and transcriptional activity with the extent of base modification. (8-oxo-dG: 8-oxo-7,8-dihydro-2′-deoxyguanosine).

Low rates of polymerase pausing are associated with a decrease in alternative RNA splicing [133,134]. Z-flipon-induced alternative splicing depends then, *in principle*, on removal of 5hC to allow the sequence to flip. Formation of Z-DNA pauses RP2 and provides time to fold the RNA and assemble a RNA splicing complex. Alternative splicing in T-Cells is associated with CTCF, which as described above, has binding sites flanked by tracts of hemi-methylated d(CpG) [127]. CTCF has greatest affinity for sites containing 5cC and binds poorly to 5mC, 5hc and 5fc [133]. Whether the effects of 5cC on CTCF-dependent alternative splicing are causal, coincidental or an effect is currently unknown. One scenario is that 5cC promotes Z-formation adjacent to the CTCF binding site. Subsequently, RP2 pauses, CTCF docks via 5cC and scans for its cognate motif. Once CTCF engages, alternative RNA splicing initiates.

Modification of 5hC to 5cC then links flipon conformation to altered RNA processing. As an analogy, Z-flipons within genes act similarly to traffic signals distributed along a highway. When lights are all set to green, traffic speed is high as there is nothing to interrupt the flow. Turning lights to red stops the traffic, enabling pedestrians to cross or cars to enter from other directions. Controlling each light with a programmable switch, like a flipon, enables optimization for particular outcomes. The most adaptable signal implementation is one in which switches are set independently. That is how an engineer would design the system. Nature probably began with flipons that were initially in sync with ALUs from the same family as they all interact with the same factors. With time, more adaptive permutations arose through natural selection.

Collectively, the different types of epigenetic modifications alter the B-Z equilibrium, changing the transcripts produced from a gene. Chemical modifications, like 8-oxo-dG and 5hC modulate the flip to 'Z' and probably affect RNA processing *in vivo*. A number of proteins also affect outcomes. Polymerase pausing due to Z-formation stalls the trailing RP2 [135], providing time for splicing and editing substrates to fold and protein complexes to assemble (figure 2*c*) [132]. Topoisomerase activity in the gene body controls transcription rate [108], potentially by limiting Z-DNA formation. TET, DNMT3A (DNA methyltransferases 3A) and repair enzymes like TDG, NEIL1 and oxoguanine glycosylase (OGG1) determine the extent and persistence of adducts, potentially regulating their

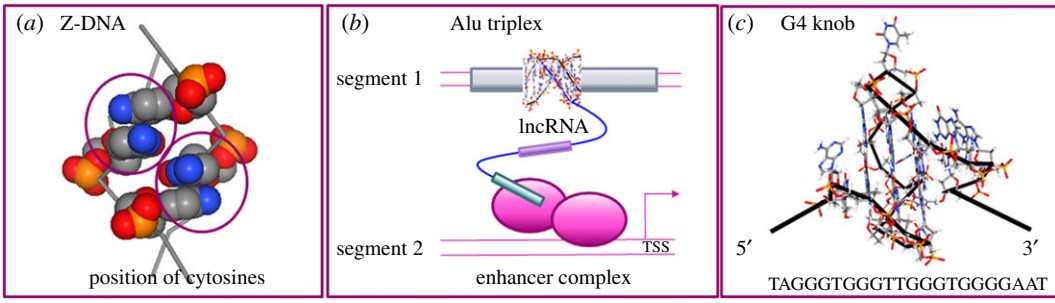

**Figure 3.** Flipon conformations. (*a*) A Z-DNA helix with the cytosines highlighted by a spacefill representation. The approximation of the C5 residues from adjacent steps is circled, and illustrates the potential for steric classes with 5fC and 5cC (blue: hydrogen, red: oxygen, grey: carbon, yellow: phosphorus) (PDB:4LB5) [126]. (*b*) A triplex bound by a lncRNA directs assembly of complexes according to the motifs present in the RNA (indicated by coloured boxes), either by local looping or through interactions with a non-contiguous DNA segment. (*c*) A G4Q formed from a single DNA strand that emphasizes its distinctive non-B-DNA structure and loops with exposed unpaired bases susceptible to modification (PDB: 2LXQ).

steady state levels. It is therefore reasonable to propose that the optimal setpoint for any particular gene in a specific tissue with respect to transcription and RNA processing is altered by a wide range of factors that impact NoBs formation. The best permutation of flipon conformations and the transcripts compiled then vary by context (figure 2*c*), changing the information extracted from their genomic neighbourhood.

How do these events relate to ALU conformation? In addition to the Z-Box [4] (figure 1*b*), ALUs contain a second potential Z-forming sequence, the XY-box (figure 1*b*, XY-box within the dotted line where 'X' over the base represents a cytosine variant and 'Y' a guanine adduct). The energetic cost of flipping the 'XY' box is Z-DNA is much higher than for the Z-Box. For the flip to occur under physiological conditions it is likely that oxidative cytosine and guanine modifications in the XY-box are required. The XY-box lies immediately 5′ to the canonical Z-Box. These two segments potentially form an extended ALU Z-DNA 'XYZ' block, with the inducible 'XY' flipon sequence acting as a sensor to signal oxidative stresses that cause base modifications (figure 1*b*). The increased length of Z-DNA might preferentially trigger DNA repair or chromatin modifications that silence the damaged region, as reported for the ADAM12 gene [136]. The 'XYZ' flip might also trigger cell-death via ZBP1 when cell-survival pathways are overwhelmed. The interplay between DNA modification and Z-DNA formation highlights an important potential regulatory role in DNA damage responses for low-complexity ALU sequences. The XYZ boxes extend the role ALUs play in innate and cancer immunity [26,84]. The widespread distribution of ALUs enables genome-wide network for surveillance of DNA damage.

What is the role of epigenetic protein modifications in flipon biology? In addition to base modifications, Z-DNA formation is associated with histone epigenetic marks that regulate gene read-out. An increase in Z-DNA formation *in vivo* occurs following trichostatin A inhibition of histone deacetylases, reflecting the boost in transcriptional activity [137]. Potential Z-DNA forming sequences overlap DNA motifs, such as the tetranucleotide d(CGCG), that are present in regions of unmethylated DNA and associated with the active histone marks H3K4me3 (histone 3 lysine 4 trimethylation) and H3K27ac (histone 3 lysine 27 acetylation) [138]. Whether flipons help set chromatin states is unknown [139]. The best possible example so far is the CSF1 promoter, whose open chromatin state depends on the BAF complex and Z-forming sequences [96]. There is a paucity of data on whether Z-flipons affect ALU chromatin in a similar manner. However, ALU DNA sequences are strongly associated with active histone marks [140], they bind over 75 transcription factors [16] and they form Z-DNA [4]. Use of ALU Z-flipons to regulate histone state and transcription factor engagement is a reasonable proposal, one that requires further exploration.

What is the interplay between B- and Z-conformations of a flipon? The B-conformation of ALU is bound in a sequence-specific manner by many proteins. For example, the TET enzymes involved in cytosine oxidations bind strong Z-DNA forming sequences as B-DNA (e.g. the TET1 motif is (A)CGCGCT [141]), with a dissociation constant in the low micromolar range [142]. Similarly, E-box binding transcription factors like MAX (MYC associated factor) also bind an alternating purine/pyrimidine motif ((C)CACGTG) capable of forming Z-DNA at NSC levels found *in vivo*. The E-box proteins bind these sequences as B-DNA with nanomolar affinity. The ADAR Zα domain binds the same sequences MAX binds with similar low nanomolar affinity, but to the Z-conformation [79]. Both

B- and Z-DNA protein binding to these sequences is affected by epigenetic modifications in the same manner. The affinity of E-boxes is higher for 5cC modification (Kd = 11 nM) than for 5hC by a factor of 10 (Kd > 110 nM) [143], while Z-formation is inhibited by 5hC and promoted by 5cC, as described above [6,126]. What then is the molecular choreography that explains how B-DNA, sequence-specific E-Box proteins and Z-DNA, structure-specific Zα domains target the same sequence? A plausible explanation is that flipons help localize transcription factors to the sequence. One possible model, but not so far experimentally addressed, involves Z-Z junctions formed by out of alternation base-pairs in the left-handed duplex. The junctions cause partial melting of the helix, producing a kink [81,144]. This distortion accentuates any base modification present. The adducts then could act as flags to initiate docking of the E-Box protein that then scans the surrounding sequence [145]. If its motif is present, the protein engages and flips the sequence back to the B-conformation. In ALU, the two Z-Z junctions are formed by an out of alternation d(GTG) sequence that separates the XY- and Z-boxes. Modification of both cytosines on the complementary d(CAC) strand will promote the flip and flag the region for the E-Box protein docking. A number of transcription factors have specificity for sequence variants of this ALU 'Z-Z' seed region [146]. If a sequence match exists, the energy stored as Z-DNA is then available to power enhancersome assembly. A dock and search strategy based on Z-flipons provides *in principle* an efficient mechanism for shrinking the search space for transcription factors to explore. It targets the cellular machinery to active chromatin.

If Z-formation is needed to assemble enhancersome in normal cells, then what leads to Z-formation prior to the initiation of transcription? Base modification due to oxidative processes or mutagens that promote flipping is one way *in principle* to lower the energetic barrier to Z-formation. Modifications by PAD peptidyl arginine deiminases is another way. These enzymes catalyse arginine deamination and citrullination of histone tails. They generate NSC by unwrapping DNA from histones cores [101]. Chromatin remodelling is a further option. In this highly dynamic process, ATP-dependent motors slide histone octamers along the DNA duplex to alter nucleosome phasing. In the process, they create nucleosome-free regions [147] and NSC that is probably sufficient for Z-formation [96]. Such chromatin reorganization is essential during development [148]. Its initiation depends upon a set of specific epigenetic marks present on the pioneer nucleosomes bound with high affinity to G-rich sequences transmitted from sperm [148–150]. In the case of the CSF1 promoter, this type of remodelling is sufficient to initiate Z-formation in cells [96]. A similar process probably accounts for the epigenetic memory associated with interferon responses [151].

In addition to changes in chromatin structure, Z-flipons alter the information content of RNA transcripts by enhancing base-modification. One example involves the p150 isoform of the ADAR. ADAR isoforms deaminate adenosine in regions of dsRNA to give inosine, which is read out as guanosine. A major subset of p150 substrates originate from chromosomal segments with AIRE that lie close to each other (less than 5000 bp separation) [19,60]. Transcripts from these regions are mostly products of RP2 [2]. They form double-stranded RNA editing substrates by folding back on themselves. *In silico* analysis reveals that editing efficiency varies with the propensity of ALU sequences to form Z-DNA [4]. Z-formation enables recognition of ALU substrates by the p150 Zα domain that binds the Z-duplex in a structure-specific, but sequence-independent manner [81,152]. This process has recently been documented in mouse prefrontal cortex [70]. Whether it is ALU Z-DNA, Z-RNA, a Z-DNA:RNA hybrid that the domain binds is still an open question as Zα binds all things 'Z'. Binding to the DNA/RNA hybrid formed during transcription potentially facilitates the handoff of ADAR from Z-DNA to Z-RNA by pausing RNA polymerases long enough for the editing substrate to finish folding and for the dsRNA binding domains to engage. By latching ADAR to its dsRNA editing substrates, the Zα domain improves the efficiency of editing by the ADAR catalytic domain [152].

Potential Z-specific substrates that do not rely on AIRE involve those with nonsynonymous edits conserved between human and mice [153]. Examples include DACT3 (R259G), a regulator of β-catenin and MEX3B (Q189R), a regulator of translation and the cyclin dependent kinase CDK13 (Q103R). Both DACT3 and MEX3B show Z-specific permanganate hypersensitivity in the Burkitt's lymphoma Raji cells, but CDK13 does not [45]. DACT3 editing along with that for CDK13 (Q103R) is reduced in mice in which ADAR is catalytically dead. It is enhanced in the spleens of normal mice where the ADAR p150 is the major isoform expressed [72]. The functional consequences of these edits require further study. In a few cases, editing by the ADAR family of enzymes can compensate for otherwise deleterious G-to-A codon mutations. [154,155]. The mutations necessitate the retention of ADAR enzymes in the genome. Here, flipons potentially compensate for maladaptive codon variants [155].

Besides acting as substrates for editing, AIRE dsRNAs also act as enhancers. An analysis of 10 650 edited sites in non-repeat sequences revealed a higher than expected presence of edited AIRE in 2 kb windows either side of the modified adenosine, more often downstream (56%) than upstream (44%) [156]. *In vitro* assays with model substrates revealed that AIRE dsRNAs increase editing efficiency of short dsRNA that would not otherwise undergo modification. One substrate analysed was NEIL1, whose editing *in vivo* is interferon-induced and almost certainly due to the ADAR p150 isoform [157]. The K242R recoding shifts enzyme substrate specificity from thymine glycol to guanidinohydantoin, a highly mutagenic oxidation product of 8-oxo-dG [158]. The ALU inverted repeat elements that enhance editing are separated by a strong Z-forming $d(AC)_n$ dinucleotide repeat sequence. Any p150-dependent editing of the transcript is probably due to either recognition of Z-DNA or a Z-DNA/Z-RNA hybrid, as the dinucleotide repeat remains single-stranded in the folded RNA editing substrate. Further study of this particular editing site will help define the important parameters for Z-dependent editing.

How does Z-dependent RNA editing affect phenotype? RNA editing impacts the proteome of a cell in many ways beyond the recoding of codons. Base modification changes secondary structure, stability, splicing and the sequence-specific pairing with other RNAs. It also impacts processing of microRNA precursors and the function of long non-coding RNAs (lncRNAs) [159]. Failure to edit dsRNAs is causal for Mendelian type I interferonopathies and arises from 'loss of function' mutations to the ADAR catalytic domain [160] and to its Zα domain [84]. The Zα P193A variant is associated with reduced dsRNA editing [98] and increased induction of the IRF3 gene [99]. The Zα mutation may also affect outcomes that are independent of editing. ADAR p150, but not p110, interacts with the RNA helicase DHX9 to suppress cRNA formation by AIRE without an increase in detectable RNA edits [161]. ADAR constructs lacking the catalytic domain target enzyme complexes involved in microRNA production that involve interactions with drosha ribonuclease III (DROSHA) and dicer 1 ribonuclease (DICER), either directly or by binding components like the DGCR8 microprocessor complex subunit [4,161–165]. ADAR p150 and p110 isoforms target different steps in miRNA processing [166,167], suggesting that some Z-flipons pattern phenotype by regulating microRNA levels. Z-flipons may explain the small size of mice in which the lethality of a catalytically dead ADAR enzyme is overcome by genetic inactivation of the type I interferon pathway [168].

# 9. T-flipons

Triplex formation is favoured in purine-rich dsDNA segments and involves docking of an additional nucleic acid strand almost always into the major groove of B-DNA. The third strand can be either purine (anti-parallel to the dsDNA purine strand, RYR (R: purine, Y: pyrimidine)) or pyrimidine (parallel to the dsDNA purine strand, YRY) [169]. Triplexes can form by intramolecular rearrangement when NSC of plasmids produce single-stranded regions that fold back onto a base-matched duplex region to form H-DNA [169]. Formation is favoured under acidic conditions (pH 4) with cytosines protonated [170]. The discussion here refers to intermolecular triplexes. These structures are only slightly underwound (11–12 bp per turn versus 10.4 bp in B-DNA) and not directly dependent on NSC for formation. Indirectly, supercoiling helps correctly align the triplex strands by increasing the flexibility of chromatin loops [171,172].

Most commonly the third triplex strand is RNA, while the duplex can be dsDNA, dsRNA or a DNA-RNA hybrid. The use of DNA as the third strand is energetically disfavoured when duplexes are made with a RNA purine strand [173]. *In vitro* triplex thermal stability depends on its base-composition and its length [174]. Formation of dC-dG:dC (':' indicates non-Watson–Crick base-pairing to the third strand) triplets is pH dependent, as it depends on protonation of cytosines on the third strand. Triplets can also incorporate non-canonical triplets such as G-C:U [175] and G-C:A [169]. In general, YRY structures are more stable than RYR as the triplets are isomorphic, stacking without distorting the helical backbone [176].

Recent studies indicate a role for triplexes in gene expression with RNA as the third strand (TSR) (figure 3*b*). Triplex formation by the MEG3 lncRNA represses TGFβ (transforming growth factor beta) pathway gene expression by targeting the polycomb repressive complex 2 to these genes. The lncRNA uses a $d(GA)_n$-rich sequence motif to bind the target genes [177]. Another triplex-forming lncRNA, PARTICLE, induces locus-specific methylation in response to low-dose irradiation [178]. In some cases, regulation is quite complex. With the sphingosine kinase 1 (SPHK1) gene, an antisense RNA KHPS1 forms a triplex at the SPHK1 enhancer, then recruits transcription factors E2F1 and p300 to initiate synthesis of eRNA-Sphk1. The eRNA displaces CTCF from its binding site, restructuring the domain for expression of SPHK1 mRNA [179]. Both YRY and RYR triplexes can form with lncRNAs,

which are non-coding RNAs that direct tissue-specific developmental programmes [180]. Whether gene silencing or RNA expression ensues depends on the non-triplex forming sequence motifs present in the lncRNA. These motifs affect the assembly and phase separation of transcription factories [30,181–183].

The ALU adenosine-rich spacer between the left and right arms of dimers also form triplexes. The variable nature of this sequence allows specific docking of a third strand (figure 1a). Genome-wide screens for triplex forming sequences reveals an enrichment for the ALU spacer motif in DNA-DNA: RNA hybrids (figure 1) [8]. The triplexes are stabilized by the interaction of the histone H3 N-terminus tail with the 2′-hydroxyl group of the RNA strand [184]. ALU RNA:DNA:DNA triplexes also involve lncRNAs. An *in silico* analysis of the NIH epigenome roadmap data reveals that 34% of triplexes formed by lncRNAs in super enhancer regions overlap SINEs [3]. Tissue-specific enhancers in these regions guide developmental programmes [185,186].

ALU T-flipons allow RNAs with low complexity sequences to regulate the read-out of genes, a model similar to that envisioned by Britten & Davidson [17]. Novel evolutionary outcomes potentially arise when an ALU T-flipons spreads to a new chromosomal location, enabling regulation of a different set of genes. Changes in the tissue-specific expression of TSRs has the potential to produce additional phenotypic variability. The gain or loss of protein-binding motifs in TSRs are another possible way to vary outcomes. The many motif combinations possible allow the assembly of new cellular machines with novel functionalities [30,181–183]. Novel TSRs probably arise rapidly as the constraints in non-coding regions are fewer than in exons. They create a medley of protein ensembles, some of which enhance survival and others with no apparent benefit.

# 10. G-flipons

Four-stranded quadruplex structures (G4) held together by a quartet of guanosines paired with both Watson–Crick and Hoogsteen hydrogen bonds are another class of flipons. G4 variants exist in which the phosphate backbones are parallel, anti-parallel or a mixture of both [51,187]. The quadruplex structures may be intermolecular (figure 1a) or intramolecular (figure 3c) with single-stranded regions connecting the quartets in various ways [188]. The *in vitro* stability depends on the presence of a metal ion at their centre. Certain metals favour parallel strand structures and while others show preference for anti-parallel folds [189]. Their functional relevance can be examined with controls unable to form G4Q. Some controls use modified residues like 7-deaza-guanosine that disrupt Hoogsteen bonding [190] and others introduce double or triple point mutations that prevent G4Q folding [187].

NSC alone is unlikely to promote quadruplex formation *in vivo*, given that other alternative NoBs form more easily [45,191]. However, transcription of linear templates *in vitro* can drive quadruplex formation in regions 5′, but not 3′ to a polymerase [192] and is due to duplex melting [193]. Enrichment for G4 forming sequences in humans is highest in the promoters and in the 5′ regions of genes and occurs at approximately equal frequency on coding and template strands [194,195]. Formation of quadruplexes in these regions is probably very rapid as it only requires refolding of a single DNA strand (figure 3c).

G4Q formation is modified *in vivo* by guanosine adducts, such as 8-oxo-dG produced by mitochondria-generated reactive oxygen species and by X-rays [122]. The modifications lower B-DNA melting temperature and facilitate G4Q formation in G-rich regions by opening up the duplex [196]. G4Q formation is favoured when there are more than four G-tracks in the neighbourhood, as the extra tracks can substitute for those damaged by oxidation [197]. About 46% of predicted G4Q forming sequences have a fifth track, including those in VEGF, c-MYC, KRAS, HIF-1α RET, HSP90, AR, PDGFR-βBCL-2 and HIF-1α promoters [197]. Damaged residues are readily accommodated in G4Q loops (figure 3c) and seed the assembly of break-excision repair (BER) complexes composed of OGG1, NEIL1, NEIL2, NEIL3 and NTH1 [197–200].

The effects of 8-oxo-dG on G4Q has been extensively studied using plasmid substrates engineered *in vitro* and tested post-transfection *in vivo*. Lesions on the template strand resolve most rapidly and downregulate gene expression by a BER-independent process [198]. Transcription stops completely when excision of 8-oxo-dG on the template strand produces an apurinic site [201]. Lesions on the coding strand are repaired more slowly and upregulate the hypoxia-induced transcription of vascular epithelial growth factor (VEGF) RNA [201,202]. Lesions to the G4Q that are enriched on the coding strand of DNA repair gene promoters also enhance gene transcription [195]. They initiate protective responses to resolve DNA damage and to reset gene expression [203]. Both processes involve chromatin remodelling. As a class, G4Q forming regions *in vivo* have epigenetic

signatures characteristic of nucleosome-depletion and elevated transcription. They shape regulatory chromatin [204].

Are there G4Q binding proteins? Crystal structures of proteins bound to G4Q indicate that many proteins recognize this conformation indirectly. The proteins either bind bases in single-stranded loops or interact with the stem formed as B-DNA transitions to G4Q [205–207]. The proteins either lock the quadruplex strands in place or dismantle it. Many of proteins the that dismantle G4Q are ATP-dependent helicases. They tug on single-stranded regions to tease the structure apart. The complementary sequence then captures the single strand to reform B-DNA, probably preventing the helicase from slipping backwards [205]. The ratchet mechanism allows helicases to slide nucleosomes along DNA, clearing the way for replication, repair, recombination and transcription complexes to assemble [205,207].

Modern genomes place G-rich sequences at promoters and use them to bootstrap development (as described above). Helicases ensure that formation of G4Q is dynamic and reversible. These enzymes enable G-flipons to switch from one conformation to another. When flipons freeze in the G4Q conformation, outcomes are often bad. 'Loss of function' helicase variants cause a wide range of monogenic disorders. The various phenotypes reflect the sequence and structural preferences of each enzyme, and the tissues that express them [208]. Their loss is compensated for by lower fidelity mechanisms that remove the G4Q blocking DNA transcription and replication [207]. These rescue pathways can also fail, leading to an increase in genome fragility. Over a longer time period, G-flipons prone to freezing are deleted from the genome [209].

Do ALUs have G-flipons? ALUs contain a single G4Q forming sequence capable of forming an intramolecular parallel-stranded quadruplex *in vitro*. The structure is made from two G-quartets stacked on each other [210]. The arrangement lacks the stability of canonical G4Q sequences made with three or more layers (figures 1*a* and 3*a*). *In vitro*, ALU wires can form by stacking on top of each other the two intramolecular G4Q made from different ALU elements [5]. Whether ALU wires form *in vivo* is uncertain, but such higher order structures *in principle* could promote nuclear organization or zip together homologous chromosomes [5]. They would be sensitive to DNA damage and destabilized by 8-oxo-dG. Alternatively, intermolecular ALU G4Q can form with quartets composed of guanosines from different ALU strands [210]. There is currently no published data confirming a biological role for ALU encoded G-flipons.

It is more likely that G-flipons are spread through the genome by elements like the primate-specific SINE-VNTR-AU (SVA) rather than by ALU alone. SVA are a subset of retrotransposons where ALU is associated with a variable number of tandem repeats (VNTR). VNTR have a repeat motif length ≥7 bp (the short tandem repeat (STR) length is 1–6 bp) [211]. VNTRs/STRs exist that are either *ab initio* and unique to humans or have a repeat length expanded when compared to apes, a total of 1584 nonredundant events. Around 92% of these human-specific repeats are associated with transposable elements. SVAs contain about half of the *ab initio* insertions and 18% of the expansions. Intronic SVAs are enriched for motifs with long homopolymer guanine and cytosine stretches with a GC content greater than 45%. They affect gene expression. For example, SVA formed *ab initio* are frequent in genes whose expression is downregulated in inhibitory neuronal cells. SVA provide evidence that the evolutionary history and biology of G-flipons differs from that of other classes.

## 11. *In vivo* evidence for flipons

The evidence for formation of non-B-DNA structures *in vivo* is strong for Z-DNA where Mendelian genetics reveal that this flipon class regulates Type I interferon responses [84]. A separate role in the initiation of inflammatory responses and necrosis is supported by mouse genetics [86–88]. Other Mendelian diseases provide evidence of biological roles for triplex and G-flipon classes, especially that subset of disorders where helicase mutations affect triplex and quadruplex resolution. Those mutations freeze flipons in the NoBs conformation [208]. Instances of Mendelian disease may also be expected from mutations to lncRNAs, but detection is challenging when multiple weak and redundant triplex interactions underlie formation of high avidity complexes [212,213]. When mutation in one triplex forming motif is partially covered by another, effects may be quantitative and carry a low risk of disease [214]. The lncRNA HOTAIR, which has five separate triplex forming sites [215], illustrates this difficulty. Obtaining a phenotype for HOTAIR by reverse mouse genetics has been challenging and controversial with findings varying by strain background [216]. Similar difficulties exist for other lncRNA, even those highly conserved between species [217]. Another approach is to

correlate expression of lncRNAs with various diseases. In cancer, elevated levels of lncRNAS, like MALAT1, THOR, SAMMSON and SChLAP1 predict lower survival [218,219]. Whether those lncRNAs are causal, coincidental or an effect of cancer is currently unresolved. Other studies show that knockout of MALAT1 reduces metastases in some tumour backgrounds [220]. In such cases, MALAT1 might compensate for 'loss of function' cancer cell mutations that prevent adaptation to new niches or derepress 'gain of function' mutations that promote spread by reducing cell attachment. Collectively the findings imply that genetic interactions are important in T-flipon biology. Mapping of phenotypes require carefully designed genetic screens.

Other evidence for non-B-DNA structures comes from genome-wide sequencing methods that use permanganate and other chemicals to fingerprint the single strand regions characteristic of each conformation [45]. For example, permanganate hypersensitive regions exist at junctions between B- and Z-DNA, one at each end of the NoBs. A limitation of these protocols is often the exclusion of repeat elements, like ALU from analysis [45], so cataloguing of all NoBs by this method is incomplete. Psoralen cross-linking of DNA reveals that many actively transcribed chromatin domains *in vivo* are both underwound and enriched for ALUs [105]. G4Q-specific reagents such as BRACO-19, pyridostatin and Phen-DC3 allow capture of quadruplexes inside cells [195]. Cross-linkable, chemically conjugated oligonucleotides demonstrate that triplexes form *in vivo* and are addressable to particular genomic targets [221]. Many other high throughput approaches, underlying much of the biology described above, have helped define flipon effects on RNA editing, epigenetic modification and genetic susceptibility. Each represents only a snapshot of a very complex picture that has multiple themes and a confounding cast of characters.

New tools under development will increase our understanding of these processes by providing time-resolved identification and imaging of non-B-DNA conformations inside intact cells. These include reagents for click chemistry at sites of transcriptionally active DNA [58], supra-resolution microscopy and fast field-resolved infrared spectroscopy [222]. These approaches enable study of proteins that modulate flipon conformation in real time and the effects of small molecule perturbagens on the processes these proteins regulate. Drugs for topoisomerases, polymerases, helicases and epigenetic modulators are already available. The low cost of whole genome sequencing will enable further Mendelian studies in disease families to map variants in hypothesized NoBs to phenotypes and help elucidate their biological functions. It is already possible to identify genotypes of interest in variant databases, but there is no phenotypic annotation available or mechanism to trace the individuals or families whose DNA was sequenced. One example concerns the Zα P193A homozygotes in humans, who are clearly viable but for whom no phenotypes are published. Reverse genetics in animal models is now easier with the DNA editing tools now available. In the future they will provide useful insights into the biologically relevant mechanisms and the genetic interactions underlying flipon phenotypes.

# 12. ALU flipons and evolution

ALU NoBs are an example of an instructive genetic code that specifies how messages are compiled. By comparison, codons represent a semantic genetic code that maps nucleotide triplets to amino acids. In the simplest form, an instructive code is binary, with the composition of a message contingent on context. Sequences that dynamically change conformation to alter genomic read-out are one way to genetically implement instructive codes. The name flipon captures their switch-like nature.

Flipons act locally. They alter the information extracted from their neighbourhood. The number of transcripts possible expands combinatorially as the flipon count in a gene increases, enhancing variability in transcript level, alternative splicing, base modification and three-dimensional RNA structure. Stochastic differences in flipons conformation within a population of cells is expected to produce a gradient of phenotypes within a tissue. Selection will optimize for flipon sets that work well enough most of the time, under most conditions, for sufficient cells, tissues or organisms to survive. Exploration and engagement of some flipon sets may not occur until something happens that compels them to compile something different. Until then, many of their transcripts will be triaged along with others that fail quality controls. Not all flipons will be useful, reflecting the random nature of their insertion. However, their exaptation may lead to new elaborations and novel protein assemblies. The medley of messages they compile will coexist with the old. There is no need to completely abandon previously successful adaptations, nor to risk irreversibly rewriting the genome by randomly mutating codons. There is a metabolic cost to the flipon strategy. It takes work to switch flipon conformations. Despite all this expenditure of energy, most of the RNA transcribed will never

be used and will require more work to dispose of it, the only gain being entropy [223]. The trade-off is a larger transcript space to explore and exploit. Quite complex systems that solve for survival can then evolve. Eventually these systems will discover ways to learn. Optimization of flipon settings is then easier than with other forms of evolution.

ALU transposition enables the spread flipon adaptations throughout the genome. Other sequences can just hitch a ride. The mash of flipons, codons, VNTRs, STRs and other motifs rapidly generates variability for natural selection to act on. This mechanism remains active in humans. Genome Wide Association Studies revealing a disproportionately high rate of recent ALU and SVA insertions at many quantitative trait loci [211,224]. ALU transposition also influences speciation. There is a 15-fold difference in ALU activity between humans and great apes [225], with evidence of positive selection in *Homo sapiens* [226,227]. The insertion of new ALUs after the human–chimpanzees split is higher in genes coding for nervous system development and function. ALU RNA editing also occurs more frequently in the human brain [228] with evidence for the positive selection of the ADAR p150 N-terminus that contains the Zα domain [229]. Along with other proposed mechanisms [230], ALUs prime the genome for rapid evolution.

# 13. Summary

ALUs have a direct impact on evolution, through the DNA damage they cause and the chromosomal rearrangements they provoke [1,231–236]. ALUs also impact evolution by the non-B-DNA structures they encode. The sequences capable of adopting these alternative conformations are called flipons.

Flipons are programmable binary switches whose conformation can be set independently of each other. They act in combination to greatly increase the coding capacity of linearly encoded genomes. Flipons act in *cis* through locally produced Z-DNA and in *trans* through triplexes formed from RNAs transcribed elsewhere. Along with G4Q that sense DNA damage, they alter the transcripts compiled. They act by changing chromatin state and phasing nucleosomes. The complex machines they assemble alter cell-specific outcomes. The cellular memories they create refashion future responses [151,237]. The Mendelian diseases they cause are instances when these processes either freeze or fail.

By their very nature, invasive ALU elements cannot help what they do. They spread by inserting themselves into active genes throughout the genome. The non-B-DNA structures they form provide an instructive code for compiling novel transcripts from wherever they land. By carelessly generating phenotypic diversity, ALUs unintentionally accelerate the evolution of many things new and unexpected.

Ethics. No human participants were involved in this study.

Data accessibility. All the data reviewed are published or available from public sources. No proprietary algorithms or material were used for this review.

Competing interests. There are no known conflicts of interest associated with this publication and there has been no significant financial support for this work that could have influenced its outcome. The author is the founder the company InsideOutBio that is committed to open science and working across disciplines. The research was conducted in the absence of any commercial or financial relationships that could be construed as a potential conflict of interest and conducted using publicly available data sources.

Funding. I received no funding for this study.

Acknowledgements. The author acknowledges the work of many scientists who prepared the data referenced in this review. I would like to thank Dr Levens and his laboratory for making available bed files with locations determined by permanganate footprinting of NoBs in the genome (https://www.ncbi.nlm.nih.gov/CBBresearch/Przytycka/software/nonbdna/nonB_DNA_predicted.tar) and Drs He, Wu and Lyu for providing bed files of their kethoxal data.

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
