## [Reviewer comments · Royal Society Open Science]

Review History

RSOS-200222.R0 (Original submission)

Review form: Reviewer 1

Is the manuscript scientifically sound in its present form?

No

Are the interpretations and conclusions justified by the results?

No

Is the language acceptable?

Yes

Do you have any ethical concerns with this paper?

No

Have you any concerns about statistical analyses in this paper?

No

Recommendation?

Major revision is needed (please make suggestions in comments)

Comments to the Author(s)

Title: ALU non-B-DNA Conformations in Evolution

ALU elements are very abundant in the human genome and they contain repetitive sequences that have the potential to form secondary DNA structures. This review article summarized the effects of Z-DNA, triplex DNA, and G4-DNA secondary structures with the potential to form in ALU elements on chromatin structure, DNA supercoiling, protein binding, RNA-DNA interaction, transcription regulation, DNA and histone modification, DNA damage and repair, and evolution.

This is a very interesting and timely article that covers a very important question in the field: what is the biological function of alternative DNA secondary structures in the human genome.

However, this reviewer has following concerns and comments:

1. As a review, this article has many speculative statements that are not completely supported by the published data cited. While speculation should be allowed in a review article, it is important to clarify which statements are based in solid "facts" and based directly on published data, and which statements may represent some hypotheses or speculation.

For example, there are quite a few places in the manuscript that contain more than one statement and the references cited only support part of the statement. Thus, the way these sentences were written might be misleading for some readers in thinking that the entire sentence was proved by citation. For example, on page 8: "The search for their cognate sequence may start with the unpaired bases at the B-Z junction [45] or at a Z-Z junction where out of alternation base pairs produce partial melting of the duplex [46]." The references are for B-Z and Z-Z junctions, but not for "search for their cognate sequence may start with the unpaired bases". Some other very important statements need references if they have direct data to support them, or the author might want to rewrite them to make it clear that these are speculations based on indirect or partial evidence. As a few examples: "Transcription factors like MAX can discover these complexes in an efficient manner by a limited scan involving facilitated diffusion along DNA" (page 8); "MAX (MYC associated factor) also bind an alternating purine/pyrimidine motif (cCACGTG) that can form Z-DNA at NSC levels found in vivo, but again these proteins bind the B-DNA conformation" (page 8); "where base-modification lowers the energetic cost of the transition sufficiently for the flip to occur under physiological conditions" (page 7); "Other chemical modifications, like 8-oxo-dG and 5hC also impact Z-DNA formation and RNA processing" (page 6); "The oxidative guanine lesions do not lead to Alu mutation." (page 5); "Base-oxidation lowers the requirement for NSC, providing an additional mechanism for signaling oxidative damage, one that enables fast responses as it only involves refolding of a single DNA strand to generate G4" (page 12)... There are others...

2. In living cells, many changes can occur together so it is important to distinguish (although it might be difficult) which alteration is a causative factor and which is a result, and which might be simply a co-incidence event. For example: in the section discussing how cytosine modification alters Z-DNA and affects transcription: "In the case of 5hC, transcription decreases when 5hC accumulates in the promoter region, but increases when 5hC accrues in the gene". Is it clear that the 5hC accumulations the reason for transcription alteration, or a bystander event, or the result? But in context, it makes the reader believe that 5hC is the reason for these changes. It needs further discussion and references.

3. Page 6: "At the promoter, high levels of 5hC likely hinders the accumulation of NSC that enables enhancer and promoter assembly at that location when stored as Z-DNA. Instead, NSC accumulates elsewhere in the gene or in some other non-B-DNA conformation". If I understand correctly, the author is saying, "in a promoter, NSC enables enhancer and promoter assembly at that location when NSC is stored as Z-DNA - in other words, if Z-DNA is formed. And high

levels of 5hC likely hinders accumulation of NSC in Z-DNA formation". Then why does high 5hC stimulate Z-DNA in gene bodies?

4. Page 9 discussed the role of Z-DNA in directing modification of RNA transcripts. Although the ADAR protein binds to Z-DNA, and dsRNA in the Z-form (Z-RNA), the RNA editing function of ADAR is not likely Z-form associated since the editing occurs on adenosine bases and converts them to inosines, which usually is not a part of Z-form. Perhaps this reviewer is misunderstanding this point, but the author should provide further clarification on this point.

5. Part of the text on page 12 is very confusing: after saying that NSC promotes G4-DNA, the author states that, "Base-oxidation lowers the requirement for NSC, providing an additional mechanism for signaling oxidative damage, Once formed, quadruplexes enhance the transcription of the DNA repair genes essential for damage resolution and chromatin remodeling." This section should be clarified.

Does it mean "Base-oxidation lowers the requirement for NSC for G4-DNA formation, which helps trigger oxidative damage signaling"? Gs in G4 structures are easier to be attacked to form 8-oxoG. But previously it was found 8-oxoGs in G runs impacted G4-DNA formation and reduced the effects of G4-DNA on polymerase stalling (Takahashi, J Am Chem Soc 2018, and Cokoi, NAR, 2018). And typically, G4-DNA reduces transcription, although it might not be the case for some genes as cited. These differences should be clarified and discussed.

Review form: Reviewer 2

Is the manuscript scientifically sound in its present form?

Yes

Are the interpretations and conclusions justified by the results?

Yes

Is the language acceptable?

No

Do you have any ethical concerns with this paper?

No

Have you any concerns about statistical analyses in this paper?

No

Recommendation?

Major revision is needed (please make suggestions in comments)

Comments to the Author(s)

The biology of unusual conformations nucleic acids – especially of DNA--are poorly studied despite the increasing current knowledge of their ubiquity in cells. Conformation diversity of DNA has been largely ignored until recently, based on the assumption that unusual conformations are odd anomalies in test tubes and would be kept infrequent in chromosomes. In this review, the author attempts to connect the biology of Alu repeats to the pool of DNA and RNA structures that are capable of forming at these highly invasive genomic elements of human genome. In the author's view, the DNA and RNA left-handed helices, three-stranded DNA-RNA hybrids, and four-stranded G-quartets emerge as key regulators of transcription, translation and genome architecture. The review highlights the potential functional roles of unusual nuclear acids conformations and calls for further exploration of these structures as functional elements in the

genome. This reviewer finds the manuscript interesting, and educating, however, in its current form, the manuscript requires significant improvements. The author needs to temper his justified enthusiasm for the field by providing a bit more discussion of the limitations of the current approaches (this will also help to direct the interested reader to critical areas that need to be solidified in order to build on them).

The main points to be clarified:

1) As discussed by the author, dimeric Alu repeats have strong nucleosome positioning at their edges where Z-DNA and quadruplex filipons are located. However, there is a strong prediction that nucleosomes and non-B DNA structures are mutually exclusive. It raises the question of how Z-DNA/quadruplex might form at Alu-repeats preloaded with nucleosomes. Possible mechanisms should be discussed.

2) Among the direct evidence cited in support of in vivo non-B DNA formation is a report describing permanganate footprinting of the single stranded regions characteristic of each conformation - citation #7. However, in that paper, Alu repeats were specifically excluded from the analysis. The author did not even mention many other papers in the field. Among them, paper based on the Z-alpha domain of ADAR for detection of Z-DNA inside the cells (for example - PMID: 27374614, 19276205). Considering the special attention given by author to ADAR function, it is unclear why the results of these papers are not presented and discussed. In addition, in recent years there have been extensive studies of quadruplex formation. This reviewer might be mistaken but it seems the results of these papers do not provide a strong support for filipon-based quadruplex biology of Alu repeats.

Minor points: there are multiple misrepresentations of the literature.

1) The author claims that "NSC also promotes quadruplex formation by unwinding B-DNA". While we cannot exclude it might happen in very rare cases, as a rule this is not generally true (PMID: 29036619, 28237796).

2) "Once formed, quadruplexes enhance the transcription of the DNA repair genes". This was not shown in the cited paper. It was just a transient transfection assay based on the SV40 promoter with quadruplex forming sequences inserted.

3) "Psoralen cross-linking of NSC reveals that actively transcribed chromatin". 1) psoralen does not crosslink supercoils, but crosslinks Watson-Crick DNA strands. 2) psoralen crosslinking is dependent on the twist of DNA, not on the writhe (writhe is a representation of DNA supercoiling).

4) "known and predicted genes across the chromosome on chromosome 21". The work was done rather on chromosome 22.

5) "NSC accumulates mostly at the TSS where topoisomerase I levels are lower". In the cited paper, the level of Top1 was much higher at TSS, but the activity was lower.

There are a couple of dozen of typos that require correction.

Decision letter (RSOS-200222.R0)

27-Mar-2020

Dear Dr Herbert,

The editors assigned to your paper ("ALU non-B-DNA Conformations in Evolution") have now received comments from reviewers.

The reviewers are enthusiastic about this review, and would like to see its publication. However, they both have a number of substantive concerns with the manuscript as it currently stands. In particular they would like to see a more careful and judicious separation of data, the conclusions that can properly be arrived at from the data, and speculation (which indeed has an important place in a review such as this). In addition, more emphasis on the limitations of the current data

and approaches would be welcome. Moreover, the bibliography could be strengthened as Reviewer 2 mentions from time to time.

We would like you to revise your paper in accordance with the referees' suggestions which can be found below (not including confidential reports to the Editor). Please note this decision does not guarantee eventual acceptance.

Please submit a copy of your revised paper before 19-Apr-2020. Please note that the revision deadline will expire at 00.00am on this date. If we do not hear from you within this time then it will be assumed that the paper has been withdrawn. In exceptional circumstances, extensions may be possible if agreed with the Editorial Office in advance. We do not allow multiple rounds of revision so we urge you to make every effort to fully address all of the comments at this stage. If deemed necessary by the Editors, your manuscript will be sent back to one or more of the original reviewers for assessment. If the original reviewers are not available, we may invite new reviewers.

- Data accessibility

<http://datadryad.org/submit?journalID=RSOS&manu=RSOS-200222>

- Competing interests

- Authors' contributions

- Acknowledgements

- Funding statement

Kind regards,
Lianne Parkhouse
Royal Society Open Science
openscience@royalsociety.org

on behalf of Professor Steve Brown (Subject Editor)
openscience@royalsociety.org

Reviewers' Comments to Author:

Reviewer: 1
Comments to the Author(s)

Title: ALU non-B-DNA Conformations in Evolution

ALU elements are very abundant in the human genome and they contain repetitive sequences that have the potential to form secondary DNA structures. This review article summarized the effects of Z-DNA, triplex DNA, and G4-DNA secondary structures with the potential to form in ALU elements on chromatin structure, DNA supercoiling, protein binding, RNA-DNA interaction, transcription regulation, DNA and histone modification, DNA damage and repair, and evolution.

This is a very interesting and timely article that covers a very important question in the field: what is the biological function of alternative DNA secondary structures in the human genome.

However, this reviewer has following concerns and comments:

1. As a review, this article has many speculative statements that are not completely supported by the published data cited. While speculation should be allowed in a review article, it is important to clarify which statements are based in solid “facts” and based directly on published data, and which statements may represent some hypotheses or speculation.

For example, there are quite a few places in the manuscript that contain more than one statement and the references cited only support part of the statement. Thus, the way these sentences were written might be misleading for some readers in thinking that the entire sentence was proved by citation. For example, on page 8: “The search for their cognate sequence may start with the unpaired bases at the B-Z junction [45] or at a Z-Z junction where out of alternation base pairs produce partial melting of the duplex [46].”. The references are for B-Z and Z-Z junctions, but not for “search for their cognate sequence may start with the unpaired bases”. Some other very important statements need references if they have direct data to support them, or the author might want to rewrite them to make it clear that these are speculations based on indirect or partial evidence. As a few examples: “Transcription factors like MAX can discover these complexes in an efficient manner by a limited scan involving facilitated diffusion along DNA” (page 8); “MAX (MYC associated factor) also bind an alternating purine/pyrimidine motif (cCACGTG) that can form Z-DNA at NSC levels found in vivo, but again these proteins bind the B-DNA conformation” (page 8); “where base-modification lowers the energetic cost of the transition sufficiently for the flip to occur under physiological conditions” (page 7); “Other chemical modifications, like 8-oxo-dG and 5hC also impact Z-DNA formation and RNA processing” (page 6); “The oxidative guanine lesions do not lead to Alu mutation.” (page 5); “Base-oxidation lowers the requirement for NSC, providing an additional mechanism for signaling oxidative damage, one that enables fast responses as it only involves refolding of a single DNA strand to generate G4” (page 12)... There are others...

2. In living cells, many changes can occur together so it is important to distinguish (although it might be difficult) which alteration is a causative factor and which is a result, and which might be simply a co-occurrence event. For example: in the section discussing how cytosine modification alters Z-DNA and affects transcription: “In the case of 5hC, transcription decreases when 5hC accumulates in the promoter region, but increases when 5hC accrues in the gene”. Is it clear that the 5hC accumulations the reason for transcription alteration, or a bystander event, or the result? But in context, it makes the reader believe that 5hC is the reason for these changes. It needs further discussion and references.

3. Page 6: “At the promoter, high levels of 5hC likely hinders the accumulation of NSC that enables enhancer and promoter assembly at that location when stored as Z-DNA. Instead, NSC accumulates elsewhere in the gene or in some other non-B-DNA conformation”. If I understand correctly, the author is saying, “in a promoter, NSC enables enhancer and promoter assembly at that location when NSC is stored as Z-DNA – in other words, if Z-DNA is formed. And high levels of 5hC likely hinders accumulation of NSC in Z-DNA formation”. Then why does high 5hC stimulate Z-DNA in gene bodies?

4. Page 9 discussed the role of Z-DNA in directing modification of RNA transcripts. Although the ADAR protein binds to Z-DNA, and dsRNA in the Z-form (Z-RNA), the RNA editing function of ADAR is not likely Z-form associated since the editing occurs on adenosine bases and converts them to inosines, which usually is not a part of Z-form. Perhaps this reviewer is misunderstanding this point, but the author should provide further clarification on this point.

5. Part of the text on page 12 is very confusing: after saying that NSC promotes G4-DNA, the author states that, “Base-oxidation lowers the requirement for NSC, providing an additional mechanism for signaling oxidative damage, Once formed, quadruplexes enhance the transcription of the DNA repair genes essential for damage resolution and chromatin remodeling.” This section should be clarified.

Does it mean “Base-oxidation lowers the requirement for NSC for G4-DNA formation, which

helps trigger oxidative damage signaling”? Gs in G4 structures are easier to be attacked to form 8-oxoG. But previously it was found 8-oxoGs in G runs impacted G4-DNA formation and reduced the effects of G4-DNA on polymerase stalling (Takahashi, J Am Chem Soc 2018, and Cokoi, NAR, 2018). And typically, G4-DNA reduces transcription, although it might not be the case for some genes as cited. These differences should be clarified and discussed.

Reviewer: 2

Comments to the Author(s)

The biology of unusual conformations nucleic acids – especially of DNA—are poorly studied despite the increasing current knowledge of their ubiquity in cells. Conformation diversity of DNA has been largely ignored until recently, based on the assumption that unusual conformations are odd anomalies in test tubes and would be kept infrequent in chromosomes. In this review, the author attempts to connect the biology of Alu repeats to the pool of DNA and RNA structures that are capable of forming at these highly invasive genomic elements of human genome. In the author’s view, the DNA and RNA left-handed helices, three-stranded DNA-RNA hybrids, and four-stranded G-quartets emerge as key regulators of transcription, translation and genome architecture. The review highlights the potential functional roles of unusual nuclear acids conformations and calls for further exploration of these structures as functional elements in the genome. This reviewer finds the manuscript interesting, and educating, however, in its current form, the manuscript requires significant improvements. The author needs to temper his justified enthusiasm for the field by providing a bit more discussion of the limitations of the current approaches (this will also help to direct the interested reader to critical areas that need to be solidified in order to build on them).

The main points to be clarified:

1) As discussed by the author, dimeric Alu repeats have strong nucleosome positioning at their edges where Z-DNA and quadruplex flipons are located. However, there is a strong prediction that nucleosomes and non-B DNA structures are mutually exclusive. It raises the question of how Z-DNA/quadruplex might form at Alu-repeats preloaded with nucleosomes. Possible mechanisms should be discussed.

2) Among the direct evidence cited in support of in vivo non-B DNA formation is a report describing permanganate footprinting of the single stranded regions characteristic of each conformation – citation #7. However, in that paper, Alu repeats were specifically excluded from the analysis. The author did not even mention many other papers in the field. Among them, paper based on the Z-alpha domain of ADAR for detection of Z-DNA inside the cells (for example - PMID: 27374614, 19276205). Considering the special attention given by author to ADAR function, it is unclear why the results of these papers are not presented and discussed. In addition, in recent years there have been extensive studies of quadruplex formation. This reviewer might be mistaken but it seems the results of these papers do not provide a strong support for flipon-based quadruplex biology of Alu repeats.

Minor points: there are multiple misrepresentations of the literature.

1) The author claims that “NSC also promotes quadruplex formation by unwinding B-DNA”. While we cannot exclude it might happen in very rare cases, as a rule this is not generally true (PMID: 29036619, 28237796).

2) “Once formed, quadruplexes enhance the transcription of the DNA repair genes”. This was not shown in the cited paper. It was just a transient transfection assay based on the SV40 promoter with quadruplex forming sequences inserted.

3) “Psoralen cross-linking of NSC reveals that actively transcribed chromatin”. 1) psoralen does not crosslink supercoils, but crosslinks Watson-Crick DNA strands. 2) psoralen crosslinking is dependent on the twist of DNA, not on the writhe (writhe is a representation of DNA supercoiling).

4) “known and predicted genes across the chromosome on chromosome 21”. The work was done rather on chromosome 22.

5) “NSC accumulates mostly at the TSS where topoisomerase I levels are lower”. In the cited paper, the level of Top1 was much higher at TSS, but the activity was lower.

There are a couple of dozen of typos that require correction.

Author's Response to Decision Letter for (RSOS-200222.R0)

See Appendix A.

RSOS-200222.R1 (Revision)

Review form: Reviewer 1

Is the manuscript scientifically sound in its present form?

No

Are the interpretations and conclusions justified by the results?

No

Is the language acceptable?

Yes

Do you have any ethical concerns with this paper?

No

Have you any concerns about statistical analyses in this paper?

No

Recommendation?

Major revision is needed (please make suggestions in comments)

Comments to the Author(s)

The author has addressed many of the previous concerns and the revised manuscript has made the things clearer and more precise. However, after clarifying the concerns, this revised manuscript still has more speculative text than fact-based descriptions. This reviewer suggests that the author inform the readers in the summary and/or at the beginning of the review, that this review article is more than a fact-based and interpretive review of the literature, and that the readers should take the conclusions using their own judgement.

Also, there are still areas where the author combines many ideas together without clear logic, which eventually made the article difficult to follow. For example (but not limited to): on page 17, the 2nd paragraph about G4Q flipons; there are many different things listed together, in the following order: 1) 8-oxo-dG destabilizes the B-DNA duplex; 2) modifications to the quartet forming guanosines also destabilize G4; 3) "excision of 8-oxo-dG from DNA produces an apurinic site that is readily accommodated in a G4-quadruplex loop" [but the paper cited is about removal of 8-oxo-dG from the coding strand by BER upregulated transcription, and suppressed transcription of 8-oxo-dG was in the template strand (Ref 125) - this citation should be used to support point #5]; 4) 8-oxo-dG induced by X-rays were centered over the center of in silico predicted G4Q [did not provide details on whether the colocalization was due to more damage or less effective repair, and did not discuss whether the damaged sequence formed G4 structures more readily or not, and did not discuss the effect of damage on G4 regions on transcription]; and finally, 5) went back to "BER repair of 8-oxo-dG associated with G4Q formation promotes

hypoxia-induced expression of the in the vascular 362 epithelial growth factor (VEGF)". --- after reading the entire paragraph, the reader still has to guess at the main points of these sentences.

Review form: Reviewer 2

Is the manuscript scientifically sound in its present form?

Yes

Are the interpretations and conclusions justified by the results?

Yes

Is the language acceptable?

Yes

Do you have any ethical concerns with this paper?

No

Have you any concerns about statistical analyses in this paper?

No

Recommendation?

Accept with minor revision (please list in comments)

Comments to the Author(s)

The manuscript is now easier to read and the author has done a nice job classifying the concepts cited results/works/ideas according to in vivo or in vitro or "in notio". However, I would suggest that the author find some other phrase than "in notio". Though the author's intent is clear, I cannot find this phrase in any on-line dictionary and a less obscure term would likely be appreciated especially by non-native speakers.

Decision letter (RSOS-200222.R1)

Dear Dr Herbert:

Thank you for submitting extensive revisions to the Manuscript ID RSOS-200222.R1 entitled "ALU non-B-DNA Conformations and Flipons in Evolution" which you submitted to Royal Society Open Science, which has now been reviewed. The comments of the reviewer(s) are included at the bottom of this letter.

While one reviewer is very positive and recommends proceeding to publication, the other reviewer still raises some concerns which I would like you to address in a second revision. Firstly, it will be important to emphasise the sometimes speculative nature and interpretative nature of the review, which of course is welcome but it would be help for you to strengthen the review by indicating where you can both at the beginning of the article and at judicious points throughout the review. In addition, please address the remaining clarifications from Reviewer 1.

Please submit a copy of your revised paper before 20-May-2020. Please note that the revision deadline will expire at 00.00am on this date. If we do not hear from you within this time then it will be assumed that the paper has been withdrawn. In exceptional circumstances, extensions may be possible if agreed with the Editorial Office in advance. We do not allow multiple rounds of revision so we urge you to make every effort to fully address all of the comments at this stage. If deemed necessary by the Editors, your manuscript will be sent back to one or more of the original reviewers for assessment. If the original reviewers are not available we may invite new reviewers.

- Ethics statement

- Data accessibility

- Competing interests

- Authors' contributions

- Acknowledgements

- Funding statement

on behalf of Dr Steve Brown (Associate Editor) and Steve Brown (Subject Editor)
openscience@royalsociety.org

Reviewer comments to Author:

Reviewer: 1

Comments to the Author(s)

The author has addressed many of the previous concerns and the revised manuscript has made the things clearer and more precise. However, after clarifying the concerns, this revised manuscript still has more speculative text than fact-based descriptions. This reviewer suggests that the author inform the readers in the summary and/or at the beginning of the review, that this review article is more than a fact-based and interpretive review of the literature, and that the readers should take the conclusions using their own judgement.

Also, there are still areas where the author combines many ideas together without clear logic, which eventually made the article difficult to follow. For example (but not limited to): on page 17, the 2nd paragraph about G4Q flippers; there are many different things listed together, in the following order: 1) 8-oxo-dG destabilizes the B-DNA duplex; 2) modifications to the quartet forming guanines also destabilize G4; 3) "excision of 8-oxo-dG from DNA produces an apurinic site that is readily accommodated in a G4-quadruplex loop" [but the paper cited is about removal of 8-oxo-dG from the coding strand by BER upregulated transcription, and suppressed transcription of 8-oxo-dG was in the template strand (Ref 125) – this citation should be used to support point #5]; 4) 8-oxo-dG induced by X-rays were centered over the center of in silico predicted G4Q [did not provide details on whether the colocalization was due to more damage or less effective repair, and did not discuss whether the damaged sequence formed G4 structures more readily or not, and did not discuss the effect of damage on G4 regions on transcription]; and finally, 5) went back to "BER repair of 8-oxo-dG associated with G4Q formation promotes

hypoxia-induced expression of the in the vascular 362 epithelial growth factor (VEGF)". --- after reading the entire paragraph, the reader still has to guess at the main points of these sentences.

Reviewer: 2

Comments to the Author(s)

The manuscript is now easier to read and the author has done a nice job classifying the concepts cited results/works/ideas according to in vivo or in vitro or "in notio". However, I would suggest that the author find some other phrase than "in notio". Though the author's intent is clear, I cannot find this phrase in any on-line dictionary and a less obscure term would likely be appreciated especially by non-native speakers.

Author's Response to Decision Letter for (RSOS-200222.R1)

See Appendix B.

Decision letter (RSOS-200222.R2)

Dear Dr Herbert,

It is a pleasure to accept your manuscript entitled "ALU non-B-DNA Conformations, Flipons and Binary Codes in Evolution" in its current form for publication in Royal Society Open Science.

Kind regards,
Anita Kritiansen
Editorial Coordinator

on behalf of Dr Steve Brown (Subject Editor)
openscience@royalsociety.org

Appendix A

Responses to Reviewer 1

Thank-you very much for your very helpful comments and the time you spent on the review.

1. As a review, this article has many speculative statements that are not completely supported by the published data cited. While speculation should be allowed in a review article, it is important to clarify which statements are based in solid “facts” and based directly on published data, and which statements may represent some hypotheses or speculation.

For example, there are quite a few places in the manuscript that contain more than one statement and the references cited only support part of the statement. Thus, the way these sentences were written might be misleading for some readers in thinking that the entire sentence was proved by citation. For example, on page 8: “The search for their cognate sequence may start with the unpaired bases at the B-Z junction [45] or at a Z-Z junction where out of alternation base pairs produce partial melting of the duplex [46].”. The references are for B-Z and Z-Z junctions, but not for “search for their cognate sequence may start with the unpaired bases”. Some other very important statements need references if they have direct data to support them, or the author might want to rewrite them to make it clear that these are speculations based on indirect or partial evidence. As a few examples: “Transcription factors like MAX can discover these complexes in an efficient manner by a limited scan involving facilitated diffusion along DNA” (page 8); “MAX (MYC associated factor) also bind an alternating purine/pyrimidine motif (cCACGTG) that can form Z-DNA at NSC levels found in vivo, but again these proteins bind the B-DNA conformation” (page 8); “where base-modification lowers the energetic cost of the transition sufficiently for the flip to occur under physiological conditions” (page 7); “Other chemical modifications, like 8-oxo-dG and 5hC also impact Z-DNA formation and RNA processing” (page 6); “The oxidative guanine lesions do not lead to Alu mutation.” (page 5); “Base-oxidation lowers the requirement for NSC, providing an additional mechanism for signaling oxidative damage, one that enables fast responses as it only involves refolding of a single DNA strand to generate G4” (page 12)...There are others...

I went through the manuscript and inserted *in vitro*, *in vivo*, *in silico* and *in notio* annotations to clarify for the reader the basis for the statements made.

I agree that the wording in the section on B-DNA binding proteins was extremely poorly written. I have extensively revised this section.

“E-boxes sequences also undergo epigenetic modifications that affect MAX binding strength. For example, the affinity of MYC for B-DNA is higher for 5cC modification ($K_d = 11\text{ nM}$) but lower for 5hC ($K_d > 110\text{ nM}$) [86]. Their effect on MYC binding is similar to the effect of Z-DNA formation. The $Z\alpha$ domain of ADAR binds those sequences in the Z-DNA conformation and with similar nanomolar affinity [48]. What is the molecular choreography that explains how B-DNA and Z-DNA binding proteins bind to the same sequence? A plausible *in notio* explanation is that flipons help localize transcription factors. One possible model, but not so far experimentally addressed, involves a Z-Z junction where an out of alternation base-pairs produce partial melting of the duplex and kinking of the helix [87]. That distortion may allow the initial docking of E-Box proteins [88], followed by sequence-specific recognition as the sequence, pulling the flipon back to the B-DNA conformation. An example of a potential Z-Z junction forming sequence is the out of alternation d(GTG) sequence that lies between the ALU canonical Z-DNA and XYZ boxes. A number of transcription factors have sequence specificity for variants of this ALU Z-Z seed region [89] and the possibility exists that the junction structure facilitates their docking. A dock and search strategy based on Z-DNA flipons *in notio* provides an efficient

mechanism for discovering their cognate sequences. If there is a match, release of the energy stored as Z-DNA is then available to power the formation of a B-DNA, sequence-specific enhancersome structures.”

2. *In living cells, many changes can occur together so it is important to distinguish (although it might be difficult) which alteration is a causative factor and which is a result, and which might be simply a co-occurrence event. For example: in the section discussing how cytosine modification alters Z-DNA and affects transcription: “In the case of 5hC, transcription decreases when 5hC accumulates in the promoter region, but increases when 5hC accrues in the gene”. Is it clear that the 5hC accumulations are the reason for transcription alteration, or a bystander event, or the result? But in context, it makes the reader believe that 5hC is the reason for these changes. It needs further discussion and references.*

These issues are addressed in the response below where it is stated

Whether the cytosine modifications are causal, coincidental or an effect is unknown.

3. *Page 6: “At the promoter, high levels of 5hC likely hinders the accumulation of NSC that enables enhancer and promoter assembly at that location when stored as Z-DNA. Instead, NSC accumulates elsewhere in the gene or in some other non-B-DNA conformation”. If I understand correctly, the author is saying, “in a promoter, NSC enables enhancer and promoter assembly at that location when NSC is stored as Z-DNA – in other words, if Z-DNA is formed. And high levels of 5hC likely hinders accumulation of NSC in Z-DNA formation”. Then why does high 5hC stimulate Z-DNA in gene bodies?*

I have rewritten this section”

“One testable explanation for the apparently contradictory effects on transcription relates to the inhibition of Z-DNA formation by 5hC. At the promoter, low levels of 5hC likely favors Z-DNA formation, potentiating its role in enhancer and promoter assembly. The high levels of 5hC in the gene body likely diminish Z-DNA formation, removing a barrier to RNA polymerase processivity and thereby increasing transcription rate. In notio, the formation of Z-DNA formation in gene bodies in vivo then depends on oxidation of 5hC to 5cC, increasing alternative RNA splicing by potentiating polymerase pausing [12, 76]. This modification then links in notio flipon conformation to altered RNA processing. In this scenario, Z-DNA flipons within genes act similarly to traffic signals distributed along a highway. When lights are all set to green, traffic speed is high as there is nothing to interrupt flow. Turning individual lights to red enables different options at those specific locations, each one independently programmable. In vivo, polymerase pausing is associated with alternative splicing [77] that varies according to 5hC and 5cC levels [76]. Whether the cytosine modifications are causal, coincidental or an effect is unknown.”

4. *Page 9 discussed the role of Z-DNA in directing modification of RNA transcripts. Although the ADAR protein binds to Z-DNA, and dsRNA in the Z-form (Z-RNA), the RNA editing function of ADAR is not likely Z-form associated since the editing occurs on adenosine bases and converts them to inosines, which usually is not a part of Z-form. Perhaps this reviewer is misunderstanding this point, but the author should provide further clarification on this point.*

I have rewritten this section:

“In addition to changes in chromatin structure, Z-DNA flipons are involved in modification of RNA transcripts. One example involves the p150 isoform of the adenosine deaminase RNA specific enzyme (ADAR). ADAR changes the information content of a transcript by deaminating adenosine in regions of double-stranded RNA (dsRNA) to give inosine, which is readout as guanosine. A major subset of substrates originate from chromosomal segments with ALU inverted repeats that lie close to each other (<5000 bp separation) [9, 95]. Transcripts from these regions are mostly products of RNA polymerase II [1] and form double-stranded RNA editing substrates by folding back on themselves. In silico analysis reveals that the extent of editing varies with the propensity of the ALU sequences to form Z-DNA [12]. Z-formation enables recognition of ALU substrates by the p150 Z α domain that binds Z-helix in a structure-specific, but sequence-independent manner [96]. Whether it is Z-DNA, Z-RNA, a Z-DNA:RNA hybrid that the domain binds is still an open question as Z α binds all things “Z”. Binding to the DNA/RNA hybrid formed during transcription potentially facilitates the handoff of ADAR from Z-DNA to Z-RNA while pausing RNA Polymerases long enough for the editing substrate to finish folding. **By latching ADAR to its dsRNA editing substrates, the Z α domain may improve the efficiency of editing by the ADAR catalytic domain [96].** Editing can impact the proteome of a cell in many ways. RNA recoding changes secondary structure, stability, splicing, codon readout and sequence-specific interactions with other RNAs, including non-coding miRNAs and long, noncoding RNAs (lncRNAs) [97]. Failure to edit dsRNAs are causal for Mendelian type I interferonopathies [51]. The Z α P193A mutation involved is associated with reduced dsRNA editing [56] and decreased induction of the Interferon Response Factor 3 (IRF3) gene [57].

5. Part of the text on page 12 is very confusing: after saying that NSC promotes G4-DNA, the author states that, “Base-oxidation lowers the requirement for NSC, providing an additional mechanism for signaling oxidative damage, Once formed, quadruplexes enhance the transcription of the DNA repair genes essential for damage resolution and chromatin remodeling.” This section should be clarified.

I have rewritten this section and included additional references (some of which were suggested by reviewer 2:

“It is unlikely that NSC alone promotes quadruplex formation of unmodified sequences given that other alternative NoBs form more easily [19, 119]. However, transcription of linear templates in vitro can drive quadruplex formation in regions 5', but not 3' to the polymerase [120]. Enrichment for G4 forming sequences in humans is also highest in the promoters and in the 5' gene regions of genes [121, 122]. It is likely that guanosine base modifications alters the rate at which quadruplexes form in these regions. One modification that has been studied extensively is 8-oxo-dG, formed in vivo from mitochondria-generated reactive oxygen species. In vitro, 8-oxo-dG destabilizes the B-DNA duplex by decreasing its melting temperature [123]. Modifications to the quartet forming guanosines also destabilize the quadruplex [124]. In vivo, excision of 8-oxo-dG from DNA produces an apurinic site that is readily accommodated in a G4-quadruplex loop [125, 126]. Since G4Q require refolding of only a single DNA strand, they can form rapidly (Fig. 3b). The effects on transcription vary with which strand bears the G4Q. Upregulation of gene expression is associated with the coding strand while 8-oxo-dG on the template strand initially decreases transcription, causing a complete stop when its excision produces an apurinic site [127]. In a genomic survey of oxidative damage induced by X-rays, the 8-oxo-dG peaks observed were centered over the center of in silico predicted G4Q [68].

There is evidence that these lesions affect transcription in cells. Both in vivo and in vitro studies suggest that base-excision repair of 8-oxo-dG associated with G4Q formation promotes hypoxia-induced expression of the in the vascular epithelial growth factor (VEGF) [127, 128]. Formation of 8-oxo-dG-induced quadruplexes increases the activity of DNA repair gene promoters assayed in vitro via transient reporter assays [122]. In such cases, quadruplex formation by G4Q-flipons signals DNA damage and potentially induces those responses necessary for their resolution and for chromatin remodeling to reset expression levels [129].”

Does it mean “Base-oxidation lowers the requirement for NSC for G4-DNA formation, which helps trigger oxidative damage signaling”? Gs in G4 structures are easier to be attacked to form 8-oxoG. But previously it was found 8-oxoGs in G runs impacted G4-DNA formation and reduced the effects of G4-DNA on polymerase stalling (Takahashi, J Am Chem Soc 2018, and Cokoi, NAR, 2018). And typically, G4-DNA reduces transcription, although it might not be the case for some genes as cited. These differences should be clarified and discussed.

These changes were included in the response to the previous question – I couldn’t locate the Cokoi, NAR 2018 reference

Responses to Reviewer 2

Thank-you for your helpful comments!

The main points to be clarified:

1) *As discussed by the author, dimeric Alu repeats have strong nucleosome positioning at their edges where Z-DNA and quadruplex flipons are located. However, there is a strong prediction that nucleosomes and non-B DNA structures are mutually exclusive. It raises the question of how Z-DNA/quadruplex might form at Alu-repeats preloaded with nucleosomes. Possible mechanisms should be discussed.*

This question is really interesting. I added the following paragraph at line 274

“If Z-formation is needed to initiate enhancersomes in normal cells, then what leads to Z-formation prior to the initiation of transcription? Base modification due to oxidative processes or by mutagens that promote flipping is one *in notio* way to lower the energetic barrier to flipping. Another way is through the chromatin remodeling that plays an essential role in development [90]. In this highly dynamic process, ATP-dependent motors remove histone octamers to phase nucleosomes [91]. Pioneer nucleosomes transmitted via sperm with high affinity for G-rich sequences and specific epigenetic marks initiate this process [90, 92, 93]. In the case of the CSF1 promoter, this remodeling is sufficient to initiate Z-formation in cells [54]. A similar process likely accounts for the epigenetic memory associated with interferon responses [94]

2) *Among the direct evidence cited in support of in vivo non-B DNA formation is a report describing permanganate footprinting of the single stranded regions characteristic of each conformation – citation #7. However, in that paper, Alu repeats were specifically excluded from the analysis. The author did not even mention many other papers in the field. Among them, paper based on the Z-alpha domain of ADAR*

for detection of Z-DNA inside the cells (for example - PMID: 27374614, 19276205). Considering the special attention given by author to ADAR function, it is unclear why the results of these papers are not presented and discussed. In addition, in recent years there have been extensive studies of quadruplex formation. This reviewer might be mistaken but it seems the results of these papers do not provide a strong support for flipon-based quadruplex biology of Alu repeats.

There are a number of different issues here that I addressed in different sections of the paper. One concern is the issue of experimental artefacts that have held the field back. I added this paragraph that explains the original decisions concerning the papers mentioned.

“Why Studying non-B-DNA structures is Hard

The study of NoBs has always been hard as experimental approaches can artefactually induce their formation from B-DNA. For example, introduction of a Z-DNA binding protein may stabilize that conformation in an unphysiological manner, or procedures that release bound histones to generate underwound DNA may promote the flip from right-handed to left-handed DNA [22]. In another instance, the binding of Z-DNA-specific antibodies to *Drosophila* polytene chromosomes is dependent on fixation conditions [23]. A similar problem potentially exists for G4Q that are remarkable stable when formed in vitro [24], meaning they can fold during the in vitro procedure designed to show their presence in vivo. The question is always the same: are these findings of relevance or just artefacts? The later view has already been written into history, at least for Z-DNA, first by the journal editors [25], then by the philosophers [26]. **In other cases, data is missing because of experimental design. It is routine to exclude repeat sequences like ALU from genome wide analyses (e.g. analysis of d(CpG) modifications removes ALU sequences by protocol [27] as do some in vivo foot printing assays [19]).** Many of the G-rich sequences lost are those expected to form non-B-DNA structures, meaning their flipon biology is not analyzed [28]. **In many experimental approaches, artefacts only become known with time. The Encode Project blacklist sequences identified for ChIP-Seq includes centromeric repeats [29]. Results based on this technique revealing the enrichment of NoBs in centromeric sequences that lack a canonical NoBs motif must therefore be interpreted cautiously [30].** The biochemical identification of NoBs binding proteins is also challenging and requires rigorous controls and structural studies to confirm findings. There have been many single paper publications in which conformation-specific proteins are claimed but without sufficient biochemical or structural studies to verify the claims [31-33]. Subsequent studies often show these proteins have a different function altogether, unrelated to NoBs. One example is Zuotin, a ribosomal co-chaperone for nascent peptides [34]. Such findings led to the proposal that Z-DNA protein binding is likely artefactual due to its recognition by lipid binding proteins [35]. Even when good reagents were available against a proven Z-DNA binding proteins like the p150 isoform of ADAR, the finding that protein location in tumor cells was cytoplasmic after interferon induction has been highly cited [36], even though nucleocytoplasmic shuttling of p150 is well documented [37-39] and other isoforms of ADAR undergo cytoplasmic localization during stress [40]. Only recently have surveys of normal tissues shown that the location of p150 is nuclear in murine thymus and spleen [41, 42], both immunological tissues. Additional challenges to identifying conformation specific non-B binding proteins also exist. Proteins like the high-mobility group family are disordered random coils in solution that only fold on binding DNA. They bind a variety of non-B structures, including triplexes and quadruplexes, none specifically [43, 44]. Another issue is that sequences capable of forming one non-B structure can also form another. An example of this are telomeric sequences that can form G4Q in vitro but in vivo may stabilize loops where the G-strand overhang forms a triplex with a telomere proximal sequence [45, 46].

There is also debate as to whether non-B structures form from DNA, RNA or a hybrid. The simple answer is that they can form from all three classes of nucleic acid duplex. The better answer, as we will discuss in notio, is that they form transiently in vivo and are involved in switching the cell from one phenotypic state to another. As such, they are non-equilibrium structures with an informational role subject to natural selection.”

I also added this reference at line 125

“ChIP-Seq based on a Z α dimer identified a total of 391 Z-forming sequences enriched in the promoter and in active histone marks such as H3K4me3 and H3K9ac [58]. The centromeric sequences found enriched in an earlier study [30] are most likely are Encode blacklist sequences [29]”.

And this to line 401

“For example, permanganate hypersensitive regions exist at junctions between B- and Z-DNA, one at each end of the sequence that has flipped to the left-handed conformation. A limitation of these protocols is often the exclusion of repeat elements, like ALU [19], so the cataloguing of all NoBs by this method is incomplete.”

BTW, I did contact Dr. Levens about this issue but received no reply.

Minor points: there are multiple misrepresentations of the literature.

1) *The author claims that “NSC also promotes quadruplex formation by unwinding B-DNA”. While we cannot exclude it might happen in very rear cases, as a rule this is not generally true (PMID: 29036619, 28237796).*

Yes, this is true – I rewrote this section to better reflect the data (starting at line 345)

“It is unlikely that NSC alone promotes quadruplex formation of unmodified sequences given that other alternative NoBs form more easily [19, 119]. However, transcription of linear templates in vitro can drive quadruplex formation in regions 5', but not 3' to the polymerase [120]. Enrichment for G4 forming sequences in humans is also highest in the promoters and in the 5' gene regions of genes [121, 122]. It is likely that guanosine base modifications alters the rate at which quadruplexes form in these regions.”

2) *“Once formed, quadruplexes enhance the transcription of the DNA repair genes”. This was not shown in the cited paper. It was just a transient transfection assay based on the SV40 promoter with quadruplex forming sequences inserted.*

I added more references and data starting at line 350

“One modification that has been studied extensively is 8-oxo-dG, formed in vivo from mitochondria-generated reactive oxygen species. In vitro, 8-oxo-dG destabilizes the B-DNA duplex by decreasing its melting temperature [123]. Modifications to the quartet forming guanosines also destabilize the quadruplex [124]. In vivo, excision of 8-oxo-dG from DNA produces an apurinic site that is readily accommodated in a G4-quadruplex loop [125, 126]. Since G4Q require refolding of only a single DNA strand, they can form rapidly (Fig. 3b). The effects on transcription vary with which strand bears the G4Q. Upregulation of gene expression is associated with the coding strand while 8-oxo-dG on the template strand initially decreases transcription, causing a complete stop when its excision produces an apurinic site [127]. In a genomic survey of oxidative damage induced by X-rays, the 8-oxo-dG peaks

observed were centered over the center of in silico predicted G4Q [68]. There is evidence that these lesions affect transcription in cells. Both in vivo and in vitro studies suggest that base-excision repair of 8-oxo-dG associated with G4Q formation promotes hypoxia-induced expression of the in the vascular epithelial growth factor (VEGF) [127, 128]. Formation of 8-oxo-dG-induced quadruplexes increases the activity of DNA repair gene promoters assayed in vitro via transient reporter assays [122]. In such cases, quadruplex formation by G4Q-flipons signals DNA damage and potentially induces those responses necessary for their resolution and for chromatin remodeling to reset expression levels [129].”

3) *“Psoralen cross-linking of NSC reveals that actively transcribed chromatin”. 1) psoralen does not crosslink supercoils, but crosslinks Watson-Crick DNA strands. 2) psoralen crosslinking is dependent on the twist of DNA, not on the writhe (writhe is a representation of DNA supercoiling).*

Understood – changed “NSC” to “DNA”

4) *“known and predicted genes across the chromosome on chromosome 21”. The work was done rather on chromosome 22.*

5) *“NSC accumulates mostly at the TSS where topoisomerase I levels are lower”. In the cited paper, the level of Top1 was much higher at TSS, but the activity was lower.*

I expanded the sentence at line 137 to read

“NSC accumulates mostly at the TSS where Z-DNA forming sequences are found [52-54]. The topoisomerase I activity here is lower (as judged by using camptothecin to trap catalytically active enzyme on DNA), even though levels of the enzyme (as judged by ChiP-seq) are much higher than in gene bodies [66].

There are a couple of dozen of typos that require correction.

These have been corrected

Appendix B

Responses to Reviewer 1

Thank-you very much for your very helpful comments and the time you spent on the review.

1. This reviewer suggests that the author inform the readers in the summary and/or at the beginning of the review, that this review article is more than a fact-based and interpretive review of the literature, and that the readers should take the conclusions using their own judgement.

I have included many statements to have help with this issue. Below are the major statements. In addition, many paragraphs mention where there is need for further experimentation to confirm the interpretations presented. I feel the discussion is now well balanced and shows that the manuscript is the work of an experimental scientist rather than a theoretical or a professional philosopher.

Abstract It is further suggested these structures enable the rapid reprogramming of cellular to offset DNA damage and regulate inflammation. The extensive experimental data supporting this form of genetic encoding is presented.

Line 72 “Here I focus on their role in the biology of ALUs, trying as we proceed to separate evidence from *in vitro*, *in vivo* and *in silico* studies from other sources. The reader is reminded that this field is rapidly evolving with many recently developed technologies paving the way to new discoveries. The discussion is intended as a guide to future experimentation.”

Line 92 “Each to their own measure of success, but the case for the biological relevance of NoBs is now very strong.”

Line 183 “Not all the necessary experiments have been done and not all the relevant data collected. What follows is an interpretation of the data we have with many of the gaps highlighted as hypotheses that require experimental evaluation.”

*2. Also, there are still areas where the author combines many ideas together without clear logic, which eventually made the article difficult to follow. For example (but not limited to): on page 17, the 2nd paragraph about G4Q flipons; there are many different things listed together, in the following order: 1) 8-oxo-dG destabilizes the B-DNA duplex; 2) modifications to the quartet forming guanines also destabilize G4; 3) “excision of 8-oxo-dG from DNA produces an apurinic site that is readily accommodated in a G4-quadruplex loop” [but the paper cited is about removal of 8-oxo-dG from the coding strand by BER upregulated transcription, and suppressed transcription of 8-oxo-dG was in the template strand (Ref 125) – this citation should be used to support point #5]; 4) 8-oxo-dG induced by X-rays were centered over the center of *in silico* predicted G4Q [did*

not provide details on whether the colocalization was due to more damage or less effective repair, and did not discuss whether the damaged sequence formed G4 structures more readily or not, and did not discuss the effect of damage on G4 regions on transcription]; and finally, 5) went back to “BER repair of 8-oxo-dG associated with G4Q formation promotes hypoxia-induced expression of the in the vascular 362 epithelial growth factor (VEGF)”. --- after reading the entire paragraph, the reader still has to guess at the main points of these sentence

I have revised the manuscript extensively and added more references to improve both the flow and to better separate the experiments so far performed from those that would be informative, if done. The rewrite of the G-flipon section is from line 536 to 573

Responses to Reviewer 2

Thank-you for your comments!

1 I would suggest that the author find some other phrase than "in notio".

“in notio” has been removed. In most cases it was replaced by the works “in principle”. I couldn’t find a substitute for “notio” that didn’t have theological, metaphysical or judicial connotations – Latin scholars were quite busy back in the day!